# Robust replication initiation from coupled homeostatic mechanisms

Mareike Berger [1] & Pieter Rein ten Wolde [1]✉

The bacterium *Escherichia coli* initiates replication once per cell cycle at a precise volume per origin and adds an on average constant volume between successive initiation events, independent of the initiation size. Yet, a molecular model that can explain these observations has been lacking. Experiments indicate that *E. coli* controls replication initiation via titration and activation of the initiator protein DnaA. Here, we study by mathematical modelling how these two mechanisms interact to generate robust replication-initiation cycles. We first show that a mechanism solely based on titration generates stable replication cycles at low growth rates, but inevitably causes premature reinitiation events at higher growth rates. In this regime, the DnaA activation switch becomes essential for stable replication initiation. Conversely, while the activation switch alone yields robust rhythms at high growth rates, titration can strongly enhance the stability of the switch at low growth rates. Our analysis thus predicts that both mechanisms together drive robust replication cycles at all growth rates. In addition, it reveals how an origin-density sensor yields adder correlations.

To maintain stable cell cycles over many generations, living cells must coordinate DNA replication with cell growth and cell division. Intriguingly, in nutrient-rich environments, the model organism *Escherichia coli* can even divide faster than the time it takes to replicate its entire chromosome[1–4]. This apparent paradox was resolved by the model of Cooper and Helmstetter in which new rounds of replication are initiated before the previous round has finished[5] (Fig. 1a). Donachie then predicted that replication is initiated at a constant volume per origin $v^*$[6]. Initiating replication at a constant origin density ensures that DNA replication is initiated once per cell cycle per origin, which is a necessary condition for maintaining stable cell cycles at all growth rates (Fig. 1a). Recent experiments at the population level showed that the average initiation volume per origin $v^*$ varies within a ~50% range over a tenfold change in the growth rate[7]. Moreover, single-cell measurements revealed that the initiation volume is one of the most tightly controlled cell-cycle parameters, varying by about 10% for any measured growth rate[3,8]. Yet, how the initiation volume is controlled so precisely, and what molecular mechanism gives rise to robust cell cycles over many generations remains despite extensive studies poorly understood[9–13].

To obtain insight into the mechanisms that control DNA replication and cell division, fluctuations in cell size have been studied[14,15].

These experiments revealed that cells obey an adder principle, which states that cells add an on average constant volume independent of the birth volume during each cell cycle. It has been proposed that cell division control is tightly coupled to the control over replication initiation[3,16,17], via a sizer on replication initiation and a timer for cell division. Yet, recent experiments revealed the existence of two adders, one on cell division and the other on replication initiation, and that these two processes are more loosely coupled than hitherto believed[8,18–23]. While these phenomenological observations are vital because they constrain any model on the molecular mechanism for initiation and cell division control, no such molecular model has yet been presented that is consistent with the experimental data.

So far, two distinct classes of models for replication initiation control have been proposed. In the here called initiator accumulation models[16,17,24–28], an initiator protein accumulates during the cell cycle proportional to the cell volume, and replication is initiated when a threshold amount per origin has accumulated. As a fixed amount of initiators per origin needs to be accumulated per replication cycle, models of this class are often seen as a mechanistic implementation of an adder[15–17,27]. Many variations of this idea with different degrees of detail have been proposed[16,25–27]. Hansen et al.[26,28] identified the initiator protein as the protein DnaA, which can be titrated away

[1]Biochemical Networks Group, Department of Information in Matter, AMOLF, 1098 XG Amsterdam, The Netherlands. ✉e-mail: tenwolde@amolf.nl

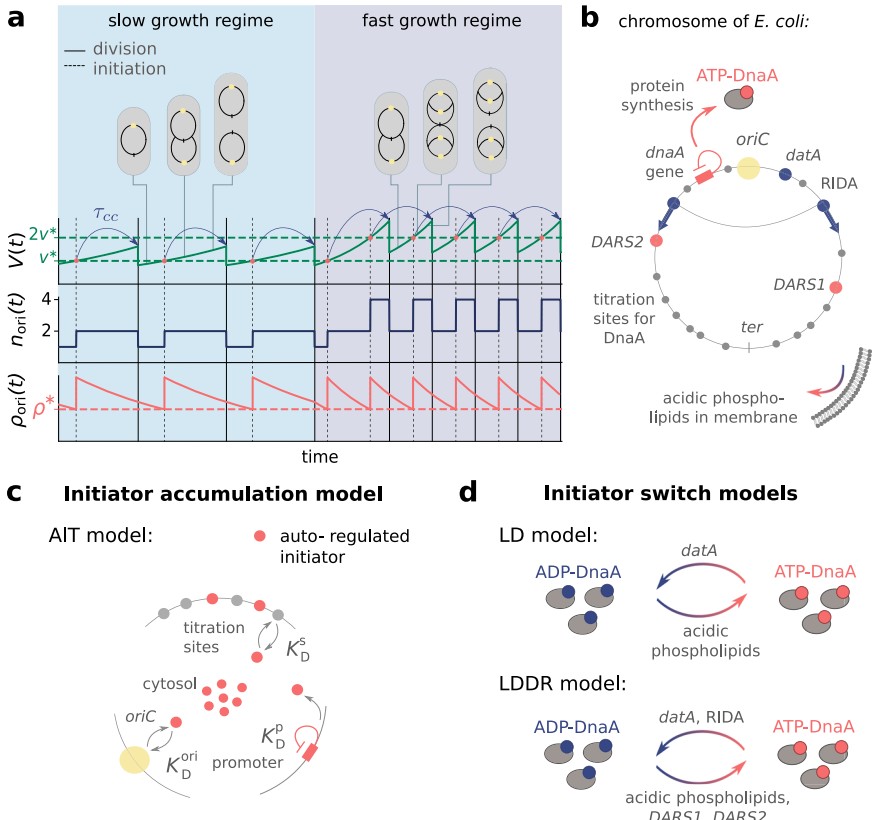

**Fig. 1 | We present two distinct models to elucidate the molecular mechanism by which *E. coli* initiates replication at an on average constant volume per origin. a** The volume $V(t)$, the number of origins $n_{ori}(t)$ and the origin density $\rho_{ori}(t) = n_{ori}(t)/V(t)$ as a function of time. Initiating replication at a constant origin density $\rho^*$ (dashed red line) or respectively a constant volume per origin $v^* = 1/\rho^*$ (dashed green line) and division a constant time $\tau_{cc}$ later (blue arrows) ensures that the cell initiates replication once per division cycle and that it maintains cell size homeostasis at slow (light blue regime) and fast (dark blue regime) growth rates. **b** Schematic representation of an *E. coli* chromosome: Replication starts at the origin (*oriC*, yellow circle) and proceeds via two replication forks to the terminus (ter, gray bar). Replication is initiated by the ATP-bound form of the initiator protein DnaA. DnaA is activated via the acidic phospholipids in the cell membrane and via the two chromosomal sites *DARS1* and *DARS2*, and deactivated via the chromosomal site *datA* and via regulatory inactivation of DnaA (RIDA), a process coupled to active DNA replication. DnaA also has a high affinity for titration sites (gray circles) located on the DNA. **c** Scheme of the AIT model: In *E. coli*, the initiator DnaA (red circles) is negatively autoregulated with the dissociation constant $K_D^p$, and can bind both to the *oriC* and the titration sites with dissociation constants $K_D^{ori}$ and $K_D^s$, respectively. **d** Scheme of the initiator switch models: In the LD model, ATP-DnaA is mainly activated via the acidic phospholipids and deactivated via the site *datA*. In the LDDR model, replication forks overlap and RIDA is the main deactivator in combination with the activators *DARS1* and *DARS2*.

from the origin by DnaA boxes, high-affinity binding sites on the chromosome[12,29]. This constant number of titration sites per chromosome sets the critical threshold number of initiator proteins required for initiating replication.

In this manuscript, we consider a mechanistic implementation of the initiator accumulation model (Fig. 1c). In *E. coli*, the initiator protein DnaA is negatively autoregulated and can be bound to titration sites on the chromosome. Following Hansen et al.[26,28], we therefore consider a model in which the initiator is autoregulated, the Autoregulated Initiator-Titration (AIT) model. While the AIT model indeed gives rise to stable cell cycles at low growth rates, it exhibits reinitiation events at high growth rates. We thus argue that the initiator titration model is not sufficient to explain the experimental data on replication initiation in *E. coli*.

The second class of models is based on a switch of the initiator protein DnaA between an active and an inactive form (Fig. 1d)[9,12,30–33]. In *E. coli*, the initiator protein DnaA forms a tight complex with ATP or ADP, but only ATP-DnaA can initiate replication by forming a complex with the chromosomal replication origin (*oriC*)[34–37]. While the total DnaA concentration is approximately constant at different growth rates[7,38], the cellular level of ATP-DnaA oscillates over the course of the cell cycle, with a peak at the time of replication initiation[33,39,40]. It has been suggested that the oscillations in the fraction of ATP-DnaA in the cell are the key to understanding how replication is regulated in *E. coli*,

but a quantitative description that is consistent with experiments is currently lacking[12,13,32,41,41–43]. Intriguingly, the level of ATP-DnaA is strictly regulated by multiple systems in the cell. DnaA is activated via acidic phospholipids in the cell membrane[44] and via two chromosomal regions called DnaA-Reactivation Sequence 1 (*DARS1*) and *DARS2*[32,39], and deactivated via the chromosomal site *datA* in a process called *datA*-dependent DnaA-ATP Hydrolysis (DDAH)[31] and via a mechanism coupled to active DNA replication, called Regulatory Inactivation of DnaA (RIDA)[33,34,45,46] (Fig. 1b). Deleting or modifying any of these systems can lead to untimely initiation, asynchrony of initiation, and changes in the initiation volume[13,31,47–51].

To dissect how these multiple mechanisms give rise to a stable cell cycle, we first study the Lipid-*DatA* (LD) model, which consists of only the acidic lipids and *datA* (Fig. 1d). This model reveals that the interplay between a constant rate of activation and a rate of deactivation that depends on the origin density gives rise to stable cell cycles. Yet, at higher growth rates these two reactions alone, based on the experimentally estimated rates of activation and deactivation, respectively, are not sufficient to generate large amplitude oscillations in the fraction of ATP-DnaA. Simulations of our Lipid-*DatA*-*DARS1/2*-RIDA (LDDR) model show that in this regime, activation via *DARS2* and deactivation via RIDA become essential.

Importantly, in our mean-field switch models, DNA replication is initiated at a threshold origin density and mechanistically they should

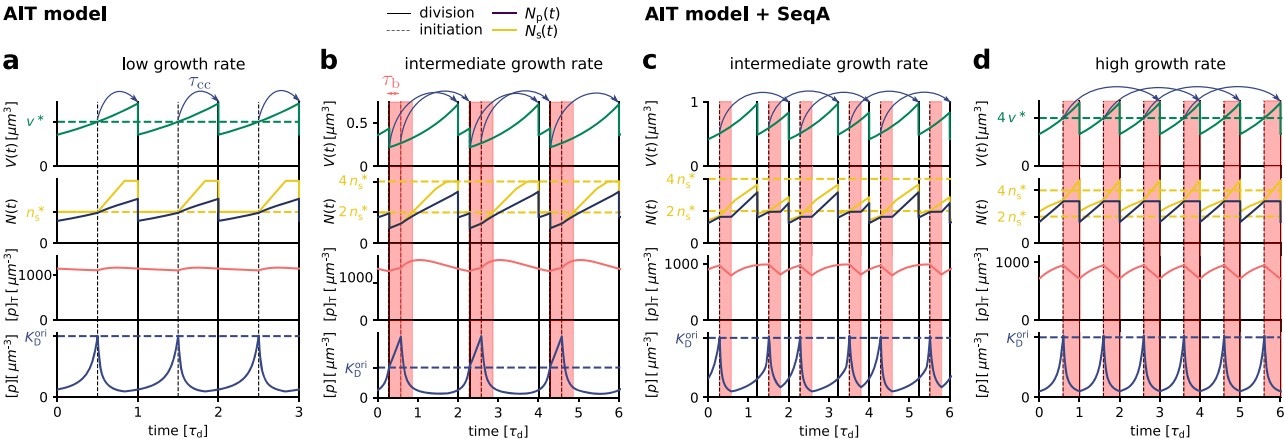

**Fig. 2 | A titration-based model can generate robust replication initiation cycles at low growth rates, but not at higher growth rates in the regime of overlapping replication forks. a–d** The volume $V(t)$, the number of initiator proteins $N_p(t)$ and titration sites $N_s(t)$, the total concentration of initiator proteins $[p]_T(t)$, and the concentration of initiator proteins in the cytoplasm $[p](t)$ as a function of time (in units of the doubling time of the cell $\tau_d$) for $\tau_d = 2$ h (**a**), $\tau_d = 35$ min (**b**, **c**) and $\tau_d = 25$ min (**d**), respectively. **a** When the number of initiator proteins per origin $n_p(t)$ exceeds the number of titration sites per origin $n_s$ (yellow dashed line), the free concentration $[p](t)$ rapidly rises to reach the threshold concentration $K_D^{ori}$ (blue dashed line), initiating a new round of replication. Due to the homogeneous distribution of titration sites on the chromosome of *E. coli* and the constant DNA replication rate, the number of titration sites then increases linearly in time. At low growth rates, new titration sites are synthesized faster than

new initiator proteins and the free concentration $[p](t)$ rapidly drops after initiation. After a fixed cycling time $\tau_{cc}$ (blue arrows) the cell divides. The initiation volume per origin $v^*$ (green dashed line) at low growth rates is constant in time. **b** However, in the regime of overlapping replication forks where the doubling time is smaller than the time to replicate the entire chromosome, $\tau_d < T_C$, new proteins are synthesized faster than new titration sites are formed. After a short period $\tau_b = 10$ min (shaded red area) during which initiation at *oriC* is blocked via the protein SeqA, replication is reinitiated prematurely, dramatically raising the variation in the initiation volume (see Fig. 5c, green line). **c**, **d** Adding SeqA, which transiently blocks DnaA synthesis for a time $\tau_b = 10$ min after replication initiation (shaded red area), prevents reinitiation events at high (**d**) but not at intermediate growth rates (**c**). (See Supplementary Table 1 for all parameters).

arguably be qualified as a sizer. Yet, we show that a stochastic version of the switch model naturally gives rise to the experimentally observed adder correlations in the initiation volume[8,18]. Fluctuations in the components that control the DnaA activation switch (lipids, HdA, Fis, IHF) are transmitted from mother to daughter cells and this generates mother-daughter correlations in the initiation volume that can explain the observed adder correlations[8].

Finally, while the AIT model inevitably fails at higher growth rates, the LDDR model is less robust at low growth rates. Yet, combining titration with the activation switch yields robust DnaA oscillations over the full range of growth rates. We thus argue that *E. coli* has evolved an elaborate set of mechanisms that act synergistically to create robust replication-initiation cycles at all growth rates.

## Results

### A titration-based mechanism is not sufficient to ensure stable cell cycles at high growth rates

Figure 1c shows the key ingredients of the AIT model. It consists of a negatively autoregulated initiator protein $p$, such that the change in copy number $N_p$ is given by

$$\frac{dN_p}{dt} = \frac{\tilde{\phi}_p^0 \lambda V}{1 + \left(\frac{[p]}{K_D^p}\right)^n} \qquad (1)$$

following the growing cell model of gene expression of Lin et al.[52] (Supplementary Note 1) with gene allocation density $\tilde{\phi}_p^0$, dissociation constant of the promoter $K_D^p$, Hill coefficient $n$ and concentration of the initiator protein $[p] = N^f/V$ in the cytoplasm. The volume $V(t)$ of the cell grows exponentially, $V(t) = V_b e^{\lambda t}$, where the growth rate $\lambda = (2)/\tau_d$, with cell-doubling time $\tau_d$, is a model parameter (Methods). The model also includes a number $N_s$ of high-affinity titration sites that are distributed randomly on the chromosome[28,53] (Methods). A new round of replication is initiated when the free initiator concentration $[p]$ reaches the dissociation constant for binding to the origin, $K_D^{ori}$. Based

on the general growth law, the cell divides a constant cycling time $\tau_{cc}$ after initiation of replication[3,4]. This choice is convenient, as it directly couples cell division to replication, thus eliminating the need for implementing an additional mechanism for cell division, yet does not affect our results, as we discuss later.

The AIT model generates stable cell cycles at low growth rates (Fig. 2a and Supplementary Fig. 3). Because the dissociation constant of the initiator protein for the titration sites $K_D^s$ is smaller than that for the origin $K_D^{ori} > K_D^s$, the cytoplasmic initiator concentration $[p]$ (Methods, Supplementary Note 2B) remains below the critical initiation threshold $K_D^{ori}$ as long as there are still unoccupied titration sites (Fig. 2a, lowest panel). Yet, when the total number of proteins $N_p$ exceeds the total number of titration sites $N_s$, the free concentration $[p]$ rapidly rises. When the free initiator concentration $[p]$ reaches the threshold $K_D^{ori}$, a new round of replication is initiated. New titration sites are now being synthesized faster than new proteins are being produced and therefore the free initiator concentration $[p]$ drops rapidly far below $K_D^{ori}$ (Fig. 2a, lowest graph). The cell then divides a constant time $\tau_{cc}$ after replication initiation, during which the volume, the number of initiator proteins, and the number of titration sites are halved. In fact, in this mean-field description cell division does not change the concentrations of the components and it therefore does not affect the replication cycle. Importantly, this mechanism ensures stable cell cycles also in the presence of *dnaA* expression noise and gives rise to the experimentally observed adder correlations in the initiation volume (Supplementary Fig. 5).

At higher growth rates, the titration mechanism, however, breaks down. Because the titration sites are homogeneously distributed over the chromosome[28,53], the rate at which new titration sites are formed after replication initiation is given by the DNA duplication rate, which is, to a good approximation, independent of the growth rate[4]. In contrast, the protein synthesis rate increases with the growth rate $\lambda$, see Eq. (1). As a result, when the system enters the regime of overlapping replication forks, where the cell division time $\tau_d$ is shorter than the time $T_C$ to replicate the DNA (Supplementary Note 2B4), the mechanism will

**Switch models (LD and LDDR):**

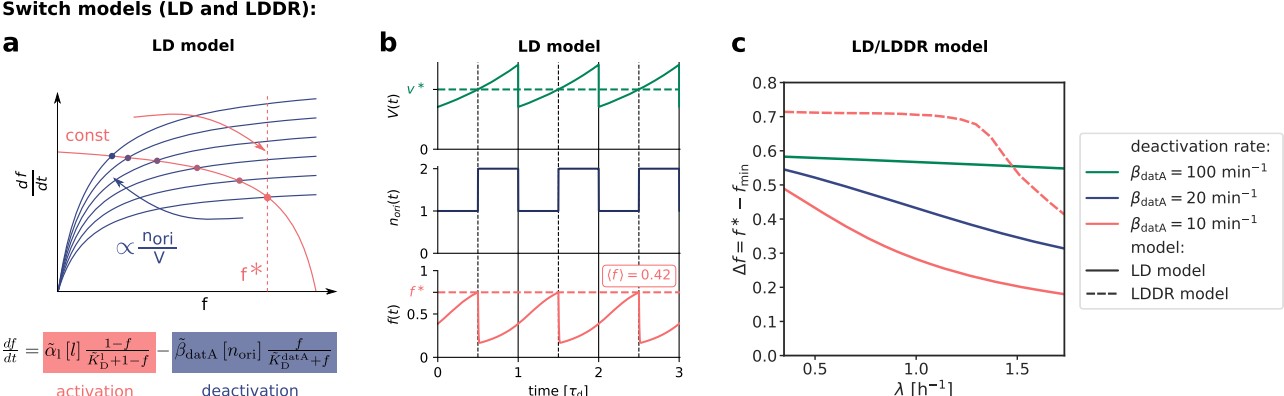

**Fig. 3 | An ultra-sensitive switch between ATP-DnaA and ADP-DnaA gives rise to stable cell cycles. a** LD model: The constant activation rate (red curve) and the origin density-dependent deactivation rate (blue curve) as a function of the active fraction of the initiator protein $f$ at different moments of the cell cycle. The steady-state active fractions are given by the intersection of the activation and deactivation rates (colored dots) and when $f$ equals the critical initiator fraction $f^*$, replication is initiated. A doubling of the number of origins leads to a decrease of the active fraction $f$. **b** LD model: The volume of the cell $V(t)$, the number of origins $n_{ori}(t)$ and the fraction of ATP-DnaA $f(t)$ from Eq. (3) as a function of time (in units of the doubling time $\tau_d = 2\,$h). The average active fraction over one cell cycle $\langle f \rangle$ is indicated in red in the third panel. Replication is initiated at a critical initiator fraction $f^*$ (red dashed line) and the system gives rise to a constant initiation volume per origin $v^*$ over time (green dashed line). **c** The amplitude $\Delta f$ of the oscillations in the active fraction $f$ as a function of the growth rate for different magnitudes of the (de) activation rates ($\alpha_l = 4.6 \times \beta_{datA}$). The amplitude of the oscillations $\Delta f$ becomes small for biologically realistic values of the (de)activation rates in the LD model (red solid curve), but not in the LDDR model (red dashed curve). (See Supplementary Table 2 for all parameters and Supplementary Fig. 9 for time traces of LDDR model).

fail to sequester the initiator after replication initiation, leading to premature reinitiation. In fact, our modeling predicts that in mutants in which $T_C$ is increased[4], the titration mechanism continues to fail in the overlapping fork regime, even though in these mutants this regime starts at lower growth rates (Supplementary Fig. 3); this supports the idea that the breakdown of the titration mechanism originates in the different scaling of the titration-site formation rate and the protein synthesis rate with the growth rate. Even when the system contains the protein SeqA, which protects the cell against immediate reinitiation events for 'an eclipse period' of about 10 min[54–56], reinitiation happens as soon as this period is over (Fig. 2b). Also, varying the number of titration sites and their affinity can not prevent premature reinitiation at high growth rates (Supplementary Fig. 3); only placing the titration sites near the origin would (Supplementary Fig. 3), but this is not consistent with experiments[28,53]. These observations show that the *E. coli* replication cycle is not regulated via titration only.

Interestingly, experiments indicate that after replication initiation SeqA not only blocks the origin, preventing immediate reinitiation, but also transiently lowers the DnaA synthesis rate[54–56]. The combination of periodic suppression of DnaA synthesis with DnaA titration enables robust DnaA rhythms at sufficiently high growth rates ($\lambda > 1.5\,$h$^{-1}$) (Fig. 2d). But at lower growth rates, corresponding to longer doubling times, the effect of SeqA becomes weaker. As a result, at intermediate growth rates ($1 < \lambda < 1.5\,$h$^{-1}$) this combination cannot prevent premature reinitiation events; the time between successive initiation events alternates between a time that is shorter than $\tau_d$ and one that is longer than $\tau_d$, giving rise to variations in the initiation volume, even though the system is deterministic (Fig. 2c). Also taking into account that the duration of the blocked period varies with the growth rate[54] does not prevent premature reinitiation events at intermediate growth rates (Supplementary Fig. 4). In this regime, another mechanism is needed.

**An ultra-sensitive switch between ATP- and ADP-DnaA gives rise to an origin-density sensor**

In the second class of models, not the total number of DnaA is the key variable that controls replication initiation, but the concentration or fraction of DnaA that is bound to ATP[30,43]. While DnaA has a high affinity for both ATP and ADP[12,57], only ATP-DnaA can initiate replication at the origin[34–36]. The switch between these two states is controlled by several mechanisms, which, we will argue, play distinct roles in different growth-rate regimes.

We first focus on the regime of slow growth in which the replication forks are non-overlapping. RIDA, a mechanism promoting ATP hydrolysis in a replication-coupled manner, becomes active upon replication initiation, but, since there are no overlapping forks, is inactive *before* replication initiation[34]. The chromosomal locus *datA* can hydrolyze ATP-DnaA via DDAH and is crucial for repressing untimely initiation events (Fig. 1b)[31]. The two chromosomal DNA regions *DARS1* and *DARS2* can regenerate ATP-DnaA from ADP-DnaA[13,32,34]. The activating site *DARS2* is reported to be only active at high growth rates[32,58,59], and the activity of *DARS1* was reported to be ten times weaker than *DARS2* in vitro[32]. In addition to *DARS1/2*, both in vitro[44,60–64] and in vivo[50,65–67] experiments indicate that acidic phospholipids can rejuvenate DnaA by promoting the exchange of ADP for ATP. Moreover, as we show in Supplementary Note 3C3, for a switch-based system, activation by *DARS1/2* is not sufficient, while lipid-mediated activation of DnaA is vital to generate stable cell cycles. In summary, our modeling in combination with experiments indicates that at slow growth, the dominant DnaA cycle of the switch setting the initiation volume consists of activation by the phospholipids and deactivation via DDAH. This cycle forms the basis of the Lipid-*DatA* (LD) model (Supplementary Note 3B).

Since the growing cell model[52] predicts that the total DnaA concentration is nearly constant in time while experiments show that it is nearly independent of the growth rate[7], we make the simplifying assumption that the total DnaA concentration is strictly constant as a function of time and the growth rate. This allows us to focus on the fraction $f = [D]_{ATP}/[D]_T$ of DnaA that is bound to ATP[68]. Exploiting that DnaA is predominantly bound to either ATP or ADP[34], the change of the active fraction $f$ in the LD model is given by

$$\frac{df}{dt} = \frac{d[D]_{ATP}}{dt}\frac{1}{[D]_T} \tag{2}$$

$$= \tilde{\alpha}_l\,[l]\,\frac{1-f}{\tilde{K}_D^l + 1 - f} - \tilde{\beta}_{datA}\,[n_{ori}]\,\frac{f}{\tilde{K}_D^{datA} + f} + \lambda(1-f) \tag{3}$$

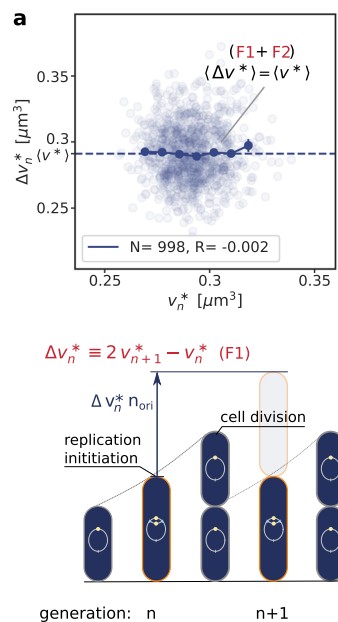

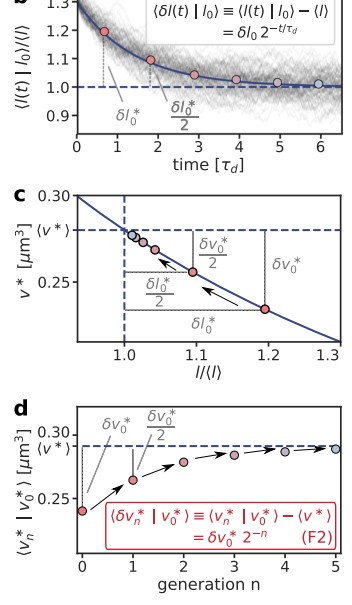

**Fig. 4 | Fluctuations in the switch components can give rise to the experimentally observed adder correlations in the initiation volume per origin $v^*$, illustrated using the LD model with lipid concentration fluctuations (Eq. (4)).**
**a** The added volume per origin between successive initiation events, $\Delta v_n^* = 2 v_{n+1}^* - v_n^*$, is independent of the initiation volume $v_n^*$ per origin and on average equal to the average initiation volume, $\langle \Delta v^* \rangle = \langle v^* \rangle$, as expected for an initiation volume adder. The cartoon below illustrates the volume $\Delta v^* n_{ori}$ that is added between successive initiation events (here, the number of origins before initiation is $n_{ori} = 1$). **b** Lipid-concentration fluctuations $l(t) \equiv [l](t)$ regress to the mean on a timescale $\tau_d = \ln(2)/\lambda$ set by the growth rate $\lambda$, such that an initial perturbation $l_0 - \langle l \rangle$ is halved every subsequent cell cycle. The thin gray lines are time traces from $N = 100$ simulations starting at an initial lipid concentration perturbation $\delta l_0$, while the solid line is the analytical prediction for the mean obtained by solving Eq. (4) subject to the same initial condition. The colored dots indicate the

average lipid concentration perturbation $\delta l$ at the moment of initiation in generation $n$ and the horizontal dashed line shows the average lipid concentration at steady state. **c** The mapping between the initiation volume $v^*$ and the normalized lipid concentration $l/\langle l \rangle$ (blue line), obtained by solving Eq. (3) in steady state in the limit of high (de)activation rates (Supplementary Note 3B1 and Supplementary Fig. 11). As in **a**, the colored dots show the average initiation volume $v^*$ as a function of $l/\langle l \rangle$ at the successive initiation events. **d** The average initiation volume (colored dots) relaxes on the same timescale $\tau_d$ as the lipid concentration, such that a perturbation $v_0^* - \langle v^* \rangle$ is halved every cell cycle, giving rise to adder correlations. In **a** the dark blue line shows the mean of the binned data and the error bars represent the standard error of the mean (SEM) per bin. The number of data points $N$ and the Pearson correlation coefficient $R$ are indicated. The model includes an eclipse period of about 10 min following replication initiation to prevent immediate reinitiation. (See Supplementary Table 2 for all parameters).

with the constant, re-normalized activation and deactivation rates $\tilde{\alpha}_l = \alpha_l/[D]_T$ and $\tilde{\beta}_{datA} = \beta_{datA}/[D]_T$ and the Michaelis–Menten constants $\tilde{K}_D^l = K_D^l/[D]_T$ and $\tilde{K}_D^{datA} = K_D^{datA}/[D]_T$. Note that because *datA* is located close to the origin, we have used here that their concentrations are equal. We further assume that the concentration of the acidic phospholipids $[l]$ is constant. The last term describes the effect of protein synthesis (Supplementary Note 3B2 and Supplementary Fig. 6). Since ATP is tenfold more abundant than ADP, new DnaA will predominantly bind ATP[34]. This term is however small at low growth rates ($\lambda \ll \tilde{\alpha}_l, \tilde{\beta}_{datA}$).

Our switch model gives rise to stable cell cycles. The crux of the model is that while the activation rate is independent of the volume of the cell, the deactivation rate decreases with the volume because it is proportional to the density of *oriC* (Fig. 3a). The ATP-DnaA fraction $f(t)$ therefore increases with increasing volume $V(t)$ as the origin density decreases (Fig. 3b). When the critical initiator fraction $f^* = [D]_{ATP}^*/[D]_T$ is reached, replication is initiated. As soon as the origin and thus the site *datA* have been replicated, the maximum of the deactivation rate doubles and the active initiator fraction $f$ decreases strongly, preventing reinitiation. As the cell continues to grow, the active initiator fraction rises again. This simple mechanism directly senses the origin density and ensures stable cell cycles (Fig. 3b).

At high (de)activation rates, the amplitude of the oscillations $\Delta f = f^* - f_{min}$ is very large (Fig. 3c). At smaller and more biologically realistic rates ($\beta_{datA} \approx 10\,min^{-1}$)[31] (Supplementary Note 3A), the amplitude of the oscillations becomes very small especially at high growth rates (Fig. 3c); this continues to hold, even when the activation-deactivation system is deeper in the zero-order regime

(Supplementary Fig. 7). Such small amplitudes do not agree with the experiments[33] and are likely to be harmful, as even small fluctuations in the active fraction could result in untimely initiation of replication.

**LDDR model with all known activators and deactivators allows for larger amplitude oscillations even at high growth rates**
Because at biologically realistic (de)activation rates the LD model fails to generate large amplitude oscillations in the active DnaA fraction at high growth rates, the question arises how the cell cycle is regulated in this regime. Interestingly, in the fast growth regime $\lambda > \ln(2)/T_C \approx 1.04/$h, where the doubling time $\tau_d$ is shorter than the time to replicate the entire chromosome $T_C$, replication is still proceeding when a new round of replication is initiated. This means that at the moment of replication initiation, the deactivation mechanism RIDA, which is associated with active replication forks, is active[46,69]. Importantly, since RIDA is a potent deactivator[48], its activity must be balanced by another activation mechanism to maintain a roughly constant initiation volume independent of the growth rate[3,4,7]. We argue that this is the principal role of *DARS2*.

We therefore included the effects of RIDA and DARS1/2 in our full Lipid-*DatA-DARS1/2*-RIDA (LDDR) model (Methods, Supplementary Note 3C). The RIDA deactivation rate is proportional to the total number of active replisomes. The activation rates of *DARS1* and *DARS2* are proportional to the copy numbers of their loci, which are located in the middle of the chromosome and are replicated at constant times after replication initiation (Supplementary Fig. 8). The LDDR model also takes into account the temporal regulation of the activities of

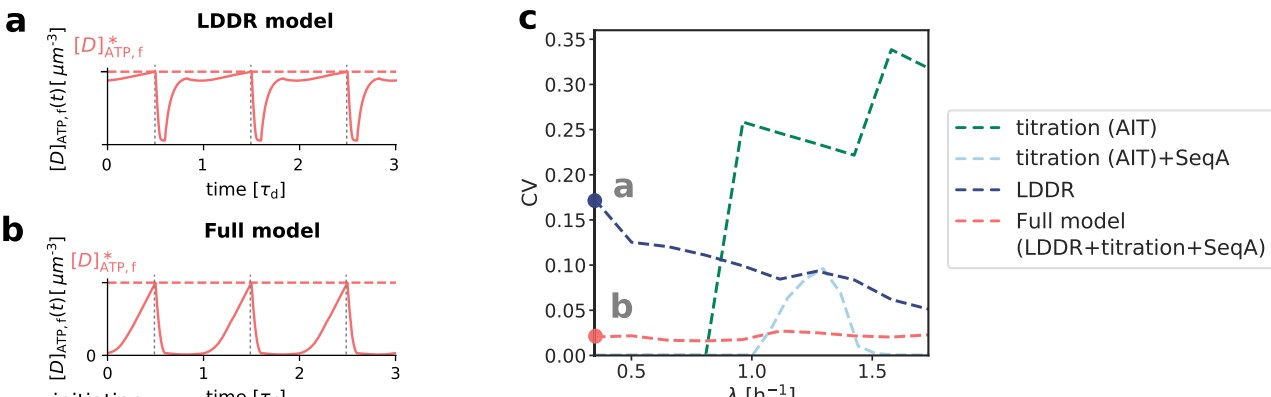

**Fig. 5 | Combining the DnaA activation switch with titration and SeqA generates robust replication-initiation cycles over a wide range of growth rates.** **a**, **b** The concentration of free ATP-DnaA $[D]_{ATP,f}(t)$ as a function of time (in units of the doubling time $\tau_d$) for $\lambda = 0.35\,h^{-1}$ as indicated in panel c. The dashed red line is the critical free ATP-DnaA concentration $[D]_{ATP,f}^*$ at which replication is initiated. While in the LDDR model the free ATP-DnaA fraction is high during a large fraction of the cell cycle (**a**, see also Supplementary Note 3C2), combining it with titration sites and SeqA gives rise to a much sharper increase of the free ATP-DnaA concentration at low growth rates (**b**). **c** The coefficient of variation CV = $\sigma/\mu$ with the standard deviation $\sigma$ and the average initiation volume $\mu = \langle v^* \rangle$ as a function of the growth rate for different models in the presence of noise in the lipid concentration.

Even in the absence of biochemical noise in DnaA synthesis, the titration model gives rise to a very high CV at high growth rates, due to premature reinitiation (Fig. 2b). Adding SeqA to the titration model can reduce the CV at high, but not at intermediate growth rates (Fig. 2c). The large coefficient of variation in the LDDR model at low growth rates is reduced significantly by the titration sites. Conversely, the LDDR model prevents the reinitiation events that inevitably occur at intermediate growth rates in the AIT+SeqA model. Combining DnaA activation with titration thus enhances the robustness of replication initiation at all growth rates, also in the presence of noise in DnaA synthesis (Supplementary Fig. 14). All models include an eclipse period of about 10 min following replication initiation to prevent immediate reinitiation[54–56]. (See Supplementary Table 2 for all parameters).

DDAH and *DARS2* via the Integrating Host Factor (IHF)[12,13,31,32] (Supplementary Fig. 8).

The LDDR model gives rise to stable cell cycles at all growth rates. Moreover, in contrast to the LD model, the LDDR model gives rise to large amplitude oscillations at all growth rates, even for realistic parameter values (Fig. 3c) (see Supplementary Fig. 9 for time traces). This is because after a new round of replication, the RIDA deactivation rate is raised immediately while the activation rates of *DARS1/2* are increased only later, after the loci have been duplicated. This differential temporal dependence of the activation and deactivation rates is key to establishing large-amplitude oscillations at all growth rates.

**A stochastic model can recover the experimentally observed adder correlations in the initiation volume per origin**

In the titration-based system, a new round of replication is initiated when the number of DnaA proteins that have been accumulated since the last initiation event equals roughly the number of titration sites, irrespective of the previous initiation volume; moreover, DnaA proteins are accumulated proportionally to the volume of the cell. These two elements together naturally give rise to adder correlations (Supplementary Note 2B7 and Supplementary Fig. 5). Yet, our switch model is a sizer at the mean-field level: replication is initiated when the origin density reaches a critical threshold. Do the experimentally observed adder correlations[8,18] rule out our switch model?

To address this question, we systematically studied the effect of fluctuations in the individual components of our switch model[70]. Consider fluctuations in the lipid concentration, modeled as

$$\frac{d[l]}{dt} = \alpha - \lambda[l] + \xi(t), \qquad (4)$$

where $\alpha$ is the production rate, the second term describes the effect of dilution set by the growth rate $\lambda$ and $\xi(t)$ models the noise resulting from protein production and partitioning upon cell division (Supplementary Note 3D). Figure 4 illustrates our findings using the LD model, but Supplementary Fig. 12 shows that the principal result also holds for the full LDDR model: the added initiation volume between consecutive

initiation events $\Delta v_n^* = 2v_{n+1}^* - v_n^*$ is indeed independent of the volume at initiation $v_n^*$, in agreement with experiments[8,18].

The concentrations of cellular components will fluctuate inevitably, and unless the components are degraded actively or produced with negative feedback control, the fluctuations will persist over several generations, regressing to the mean on a timescale set by the growth rate (Fig. 4b). The components that control the threshold of the DnaA activation switch are no exception to this rule. Moreover, their concentration fluctuations will give rise to fluctuations in the initiation volume $v^*$ (Fig. 4c) that, to a good approximation, relax on the same timescale because (de)activation is fast compared to the growth rate and the mapping between these components and the initiation volume is roughly linear. If this timescale is set by the growth rate, then deviations of $v^*$ from its mean are on average halved every cell cycle (Fig. 4d), and this gives rise to adder correlations (Methods, Supplementary Note 3D)[8]. Fluctuations in switch components that relax with the growth rate, be they lipids or proteins that modulate the activity of *datA*, RIDA, or *DARS1/2* like IHF and Hda[12,13,31,32,46], thus give rise to adder correlations (Supplementary Fig. 13).

**Coupling titration with DnaA activation enhances robustness**

All our systems are stable in the presence of biochemical noise. The concentrations do not diverge, also not in the titration-based system at high growth rates (Fig. 2). Yet, the precision of replication initiation differs markedly between the respective models, see Fig. 5. The protein synthesis and the titration-site formation rate scale differently with the growth rate, which means that a titration-based mechanism inevitably breaks down at sufficiently high growth rates, causing premature reinitiation events and a dramatic rise of the coefficient of variation (CV) in the initiation volume; even in the absence of any biochemical noise, the CV becomes larger than that reported experimentally[3,8] (Fig. 5c). The transient suppression of DnaA synthesis by SeqA after replication initiation can prevent these premature reinitiation events, but only at high growth rates: at intermediate growth rates, the CV of a system based on only titration and SeqA still rises strongly. This indicates that the activation switch is essential (Fig. 5 c). But could it be sufficient? Our modeling predicts it could because the LDDR model can generate robust oscillations at all

growth rates. Yet, our modeling also predicts that titration helps the switch by shaping the oscillations in the free concentration of ATP-bound DnaA (Fig. 5a, b, Methods), such that the precision of replication initiation in the presence of noise is significantly enhanced (Fig. 5c). In Supplementary Note 4A2 we show that a concentration cycle, as generated by titration and SeqA, can generically enhance an activation cycle, as driven by the switch, by increasing the steepness of the oscillations; this tames the propagation of fluctuations in the free concentration of active DnaA to the initiation volume[71] (Supplementary Fig. 15). Combining the switch with titration can thus protect the system against fluctuations in the switch components.

## Discussion

While the two mechanisms of titration and protein activation have so far been mostly studied independently[8,28,31–33,38,39], our manuscript indicates that the robustness arises from the coupling of the two. Interestingly, recent experiments, which show that replication is neither controlled by titration only nor by a DnaA activation switch only, support this prediction from our model[72]. Moreover, the idea that coupling an oscillation in the concentration with an oscillation in the fraction gives rise to more robust rhythms than either oscillation alone, is very generic. Our results are thus expected to apply to any cell-cycle control system that combines titration with protein activation or modification. This finding is of particular interest given the recent observation that also higher organisms employ not only protein modification but also titration for cell-cycle control[73,74]. In fact, the evidence is accumulating that also other oscillatory systems, most notably circadian clocks in cyanobacteria and higher organisms, derive their robustness to changes in the growth rate by intertwining a protein modification cycle with a protein concentration cycle[75–79].

The mechanisms of titration and activation belong to distinct classes of replication initiation control. The titration-based AIT model is an example of an initiator accumulation model, in which an initiator protein needs to accumulate to a threshold number to initiate replication[4,8,25,28,38]. In contrast, the DnaA activation switch is an example of a push-pull network in which the regulator switches between an inactive and an active state. Conceptually, this switch model is different from the accumulation model because replication is triggered at a critical concentration or fraction and not at a critical number of accumulated initiator proteins. In the switch model, the concentration of ATP-DnaA is set by the balance between DnaA activation and deactivation. Because the (de)activation rates depend on the origin density, the critical initiator concentration maps onto a critical origin density for replication initiation. This switch system is thus a bonafide origin-density sensor.

In recent years, single-cell tracking data have revealed that not only *E. coli* but also other evolutionary divergent organisms like *Bacillus subtilis*[15], *Caulobacter crescentus*[14], the archaeon *Halobacterium salinarum*[80], and even budding yeast[81], obey a division adder principle. Our study gives a new perspective on the question whether a cell cycle is controlled via a sizer or adder. While the titration mechanism naturally qualifies as an adder, our switch model should be characterized as a sizer at the mean-field level: the mechanism is based on sensing the origin density. Yet, the inevitable fluctuations in the components that control the density threshold for replication give rise to adder correlations. This idea is general and likely applies to other organisms that obey the adder principle: adder behavior may result from size sensing. Our prediction could be tested by measuring the critical active DnaA concentration for replication initiation and how its fluctuations relax. Since ATP binding induces a conformational switch of DnaA[82], developing a FRET-based ATP-DnaA sensor may be feasible.

While our models are built on a wealth of data, they all make the simplifying assumption that the cell divides a constant time $\tau_{cc}$ after

replication initiation, independent of the growth rate. Experiments indicate, however, that this is an oversimplification[3,8,18,21–23,83] and that cell division is more loosely coupled to replication initiation[8,18]. Importantly, our results on replication initiation control are robust to the assumption of a constant $\tau_{cc}$, because on average cell division does not change the densities of the components. Indeed, while this assumption will affect the correlations between the cell volume at birth and the initiation volume, it does not change the correlations between the initiation volume and the volume added until the next initiation event (Supplementary Fig. 20).

Our model is supported by many experimental observations. Of particular interest are mutants in which the (de)activation mechanisms are modified or even deleted, because these allow us to test the prediction that replication initiation is controlled by the activation switch (Supplementary Note 4B1). Naturally, our model can reproduce the observations on which it is built: deleting *datA*[31,40,84] and deactivating RIDA[31,33,34,45,85] raises the active fraction of DnaA, while deleting *DARS1/2*[32,39] reduces it (Supplementary Fig. 17). Our model then predicts that impeding activation increases the average volume per origin, while weakening deactivation has the opposite effects. Many experiments support these predictions: deleting *DARS1* and/or *DARS2* increases the initiation volume per origin[39,86], while deleting *datA* decreases it[86–88]. Our model cannot only reproduce these observations, but also the effect of combinations of deletions of these chromosomal loci on the initiation volume (Supplementary Fig. 17). Moreover, it can describe how the initiation volume per origin changes when *datA* or *DARS2* is translocated towards the terminus[84,89,90] (Supplementary Fig. 17). In addition, our model can reproduce the observation that increasing the number of titration sites via multicopy plasmids increases the initiation volume per origin[91] (Supplementary Note 4B2, Supplementary Fig. 16), while increasing the DnaA concentration reduces it[4,72,92,93] (Supplementary Note 4B3, Supplementary Fig. 18). Taken together, these experiments support the idea that replication initiation is controlled by both titration and DnaA activation.

Intriguingly, the relative position of *DARS2* with respect to the origin and the terminus is conserved in various genomes of different sizes and strains[86], suggesting it plays an important role. Our modeling provides the following rationale: In the high growth-rate regime of overlapping replication forks, *DARS2* not only serves to balance the strong deactivation by RIDA to yield a roughly constant initiation volume, but also needs to generate oscillations in concert with RIDA. Because the activities of both *DARS2* and RIDA are proportional to the origin density, *DARS2* can only play this dual role if its position meets two constraints: On the one hand, the activity of *DARS2* should rise as late as possible in order to push the active initiator fraction down right after initiation. On the other hand, to achieve a nearly constant initiation volume independent of the growth rate, the activity of *DARS2* must be high to counteract RIDA before the next initiation event; indeed, moving *DARS2* towards the terminus increases the initiation volume[89,90] (Supplementary Fig. 17i). The shortest period until replication is set by the highest doubling time of *E. coli*, $\tau_d \approx 18$ min. The position of *DARS2* in the middle of the chromosome ($\tau_{d2} \approx 16$ min) therefore naturally results from our model.

Arguably the most enigmatic element of our model is the role of the lipids in rejuvenating DnaA. In vitro experiments have shown that acidic phospholipids in the cell membrane promote dissociation of nucleotides from DnaA very effectively[44], and can restore replication activity of DnaA bound to ADP[60,61]. Depleting acidic phospholipids in vivo can lead to growth arrest[50] and inhibit initiation at *oriC*[65]. These experiments support the idea that lipids can reactivate DnaA by promoting the exchange of bound ADP for ATP. On the other hand, it has been observed that the lethal effect of a *pgsA* null mutation, which causes a complete lack of the major acidic phospholipids, is alleviated by mutations that change the membrane

structure[94,95]. More recently, it has been reported that while down-regulating *pgsA* reduced the growth rate, the initiation volume was not significantly altered[96]. We have therefore also studied models in which lipid-mediated DnaA is absent (Supplementary Note 5A). Our modeling predicts that lipid-mediated DnaA activation is essential for the switch (Supplementary Fig. 21a–d). The capacity of the switch to act as an origin-density sensor hinges on the idea that the activation and deactivation rates scale differently with the origin density. Without the lipids, only protein synthesis remains as an activation mechanism that does not scale with the origin density (Eq. (3)). Consequently, to obtain a switch-based system that is even deterministically stable, the rates of all other (de)activation mechanisms must be comparable to or smaller than the growth rate. This dramatically lowers the amplitude of the oscillations, to such a degree that the system likely becomes unstable in the presence of biochemical noise. The full model, which combines the switch with titration and SeqA, is, however, surprisingly resilient to the removal of lipids, although the latter does compromise the precision of replication initiation (Supplementary Fig. 21e–g). Indeed, while a system built on only a titration-based concentration cycle fails at high growth rates, and a system based on only an activation cycle driven by DDAH, RIDA and *DARS1/2* is likely unstable at all growth rates, the system that combines these cycles is able to generate strong rhythms at all growth rates (Supplementary Fig. 21e–g). Even an activation cycle without lipids can thus interact with a concentration cycle to drive robust replication cycles. The full model without the lipids also exhibits adder correlations (Supplementary Fig. 21h).

It has also been suggested that lipid-based DnaA rejuvenation is contingent on *oriC*[60] (Supplementary Note 5B). However, a lipid-mediated DnaA activation rate that scales with the origin density effectively reduces the *datA*-mediated deactivation rate; this yields a switch that behaves similarly to that of the lipid-devoid system, because protein synthesis is again the only DnaA activation mechanism that is independent of the origin density. In summary, lipids enhance replication initiation, but only if their effect is independent of the origin density.

Perhaps the most non-trivial prediction of our model is that the relaxation timescale of the switch components governs whether the switch generates adder or sizer correlations in the inter-initiation volume. The experiments of Si et al. provide strong support for this prediction: by expressing DnaA in an oscillatory fashion, the adder is turned into a sizer[8], precisely as our model predicts (Supplementary Fig. 19).

Our modeling predicts that negative autoregulation does not play a direct role in replication initiation. This is supported by recent experiments, which show that the average initiation volume and precision of replication initiation are only weakly affected in strains with constitutive *dnaA* expression[72]. Following Hansen et al.[38], we believe that negative autoregulation only plays an indirect role, by setting the growth-rate dependence of the DnaA concentration. Experiments have revealed that the total DnaA concentration varies with the growth rate, anticorrelating with the initiation volume[7]. However, the variation of both the total DnaA concentration and the initiation volume is rather weak, i.e. about 50% over a tenfold change of the growth rate[7]. It seems likely that negative autoregulation is crucial for constraining the growth-rate dependence of the total DnaA concentration[97,98] and hence the initiation volume[3,4]. How negative autoregulation with a differential sensitivity of the DnaA promoter to DnaA-ATP and DnaA-ADP[68,99] and titration conspire to shape the growth-rate dependence of the DnaA concentration and the initiation volume, we leave for future work.

Another open question remains why *E. coli* has evolved two different switch systems, Lipid-*DatA* (LD) and *DARS1/2*-RIDA (DR). In principle, a switch based on activating lipids and deactivating *datA* would be sufficient to control replication initiation at all growth rates.

Yet, to ensure high amplitude oscillations in the active DnaA fraction at high growth rates, the (de)activation rates would have to be higher than observed (Fig. 3c). This would require higher turnover rates of ATP, which may not be achievable when the growth rate is low. Our model thus suggests that *E. coli* has evolved a slow system to control the initiation volume at low growth rates, the lipids-*datA* system, and then switches on a faster, more energy-consuming system at higher growth rates, based on RIDA and *DARS2*.

Finally, our model predicts that in the regime of non-overlapping replication forks it should be possible to move the system from a switch-dominated regime to a titration-based one by increasing the number of titration sites or decreasing the basal synthesis rate of DnaA. Our model predicts that the dependence of the initiation volume on the number of titration sites or basal synthesis rate exhibits a marked, characteristic crossover when the system transitions between these two regimes (Supplementary Fig. 16). This is a strong prediction that could be tested experimentally.

## Methods
### Cell cycle simulations
Our molecular models of replication initiation are based on coupled differential equations that are propagated in time. In all models, the volume grows exponentially with a growth rate $\lambda$ which is a cell-cycle independent model parameter. Each simulation starts with one fully replicated chromosome and a new round of replication is initiated when the for initiation relevant quantity (which depends on the details of the respective model as explained below) attains a critical threshold. New rounds of replication then start deterministically at all origins and the replication forks advance with a constant replication rate $1/T_C$. Cell division is triggered a fixed cycling time $\tau_{cc} = T_C + T_D$ after replication initiation. This choice ensures that each replication termination event is followed by a division event; in Supplementary Note 4B5 we also study other model variants. Upon cell division, the cell volume, the number of chromosomes, and the number of proteins are divided by two. At each cell division event, one of the two daughter cells is kept at random and the other one is discarded. We thus obtain cell lines whose statistics can then be analyzed. The cell cycle simulations were performed in python 3 (version 3.7.3) and the data was analyzed using jupyter notebooks (version 6.0.0). The code is publically available (see Code Availability section).

### Autoregulated Initiator-Titration (AIT) model
We use the recently developed growing cell model of gene expression proposed by Lin et al.[52] to model the change in the total number of initiators over the course of the cell cycle (Eq. (1), Supplementary Note 1). In this model, contrary to arguably more traditional models of gene expression[97,100–103], transcription is limited by the availability of RNAPs while translation is limited by the ribosomes. This gives rise to mRNA and protein copy numbers that are proportional to the cell volume, as recent experiments indicate[52,104–109]. Importantly, the protein synthesis rate is, as observed very recently[104], proportional to the volume, which is a crucial requirement for the stability of the initiator accumulation model (Supplementary Note 2A). The total initiator concentration is then obtained by dividing the total number of initiators by the volume of the cell. Each chromosome contains a constant number of homogeneously distributed titration sites and the initiator proteins can be either freely diffusing in the cytoplasm or bound to these titration sites. As binding and unbinding dynamics of the initiator protein to the titration sites are relatively fast[110–112], we assume for simplicity a quasi-equilibrium state and use a quadratic equation to calculate at every given total titration site concentration and total initiator concentration in a cell the concentration of initiators freely diffusing in the cytoplasm (Supplementary Note 2B2). When a critical free initiator concentration is reached, replication is initiated and the number of titration sites $N_s(t)$ increases linearly from the moment of

initiation of replication $t_i$ until the end of replication at $t_i + T_C$:

$$N_s(t) = \begin{cases} N_0 & \text{for } t < t_i \\ N_0 + N_0 \frac{t - t_i}{T_C} & \text{for } t_i \leq t < t_i + T_C \\ 2N_0 & \text{for } t \geq t_i + T_C \end{cases} \qquad (5)$$

with the C-period $T_C \approx 40$ min being the time to replicate the entire chromosome and $N_0 = n_s n_{ori}$ is the total number of titration sites before replication initiation, given by the number $n_s$ of titration sites per chromosomes times the number $n_{ori}$ of origins before replication initiation.

### Activation switch models

As the total DnaA concentration $[D]_T$ is to a good approximation constant in time and as a function of the growth rate[7,52], in the activation switch models we assume that $[D]_T$ is constant and only evolve the ATP-DnaA fraction $f(t)$ in time. At a critical ATP-DnaA fraction $f$ replication is initiated. The LDDR model contains all known (de)activators with their temporal regulation over the course of the cell cycle: the number of catalytic RIDA complexes is proportional to the number of origins with a rate $\beta_{rida}$[46,113] that is only non-zero during the period of active replication $T_C$ (Supplementary Fig. 8b). The chromosomal sites *DARS1* and *DARS2* are located near the middle of the chromosome and are replicated at constant times $\tau_{d1}$ and $\tau_{d2}$, respectively, after the origin (Supplementary Fig. 8a). The activities of DDAH and *DARS2* are temporally regulated during the cell cycle via binding of the integrating host factor (IHF)[12,13,31,32]. We model these observations via step functions $\alpha_{d2}(t - t_i)$ and $\beta_{datA}(t - t_i)$ with a high and a low rate for *DARS2* and DDAH, respectively, that vary as a function of the time since initiation of replication $t - t_i$ (Supplementary Fig. 8b). *DARS1* activation is modeled via a constant activation rate $\alpha_{d1}$. Like in the LD model, we assume that every newly synthesized DnaA binds ATP rather than ADP right after synthesis (Supplementary Note 3B2). Summing up, we obtain the following change in the ATP-DnaA fraction in the LDDR model (Supplementary Note 2C1):

$$\begin{aligned} \frac{df}{dt} = &\left( \tilde{\alpha}_l [l] + \tilde{\alpha}_{d1} [n_{ori}(t - \tau_{d1})] \right. \\ &\left. + \tilde{\alpha}_{d2}(t) [n_{ori}(t - \tau_{d2})] \right) \frac{1 - f}{\tilde{K}_D + 1 - f} \\ &- \left( \tilde{\beta}_{datA}(t) + \tilde{\beta}_{rida}(t) \right) [n_{ori}] \frac{f}{\tilde{K}_D + f} + \lambda(1 - f) \end{aligned} \qquad (6)$$

with the re-normalized activation and deactivation rates $\tilde{\alpha}_l = \alpha_l / [D]_T$, $\tilde{\alpha}_{d1} = \alpha_{d1} / [D]_T$, $\tilde{\alpha}_{d2} = \alpha_{d2} / [D]_T$, $\tilde{\beta}_{datA} = \beta_{datA} / [D]_T$ and $\tilde{\beta}_{rida} = \beta_{rida} / [D]_T$ and the Michaelis-Menten constant $\tilde{K}_D = K_D / [D]_T$. The parameters are described in the Supplementary Note 3A and their values are listed in Supplementary Table 2.

### Derivation of adder correlation from size sensor

Lipid concentration fluctuations $l(t) \equiv [l](t)$ modeled according to Eq. (4) regress to the mean $\langle l \rangle$ on a timescale given by the cell-doubling time $\tau_d = \ln(2)/\lambda$. The average deviation of the lipid concentration from its mean $\langle \delta l(t)|l_0 \rangle \equiv \langle l(t)|l_0 \rangle - \langle l \rangle$ subject to an initial concentration fluctuation $\delta l_0 = l_0 - \langle l \rangle$ is given by:

$$\langle \delta l(t)|l_0 \rangle = \delta l_0 2^{-t/\tau_d}. \qquad (7)$$

Assuming fast (de)activation compared to the growth rate and exploiting that the mapping $v^*([l])$ is roughly linear (Fig. 4c), the decay of lipid fluctuations causes the initiation volume to regress to the mean on the timescale of the doubling time $\tau_d$ (Fig. 4d):

$$\langle \delta v_n^*|v_0^* \rangle = \delta v_0^* 2^{-n}, \qquad (8)$$

where $\langle \delta v_n^*|v_0^* \rangle \equiv \langle v_n^*|v_0^* \rangle - \langle v^* \rangle$ is the average deviation from the average initiation volume $\langle v^* \rangle$ given an initial initiation volume $v_0^*$ after $n$ cell cycles. Thus, fluctuations in the initiation volume relax to the mean via a geometric series, akin to that observed for the volume at birth[15]. Combining $\langle \delta v_n^*|v_0^* \rangle = \delta v_0^* 2^{-n}$ with the definition of the added initiation volume $\Delta v^* \equiv 2v_{n+1}^* - v_n^*$ (Fig. 4a) shows that the average added initiation volume is independent of the initiation volume, and equal to the average initiation volume $\langle \Delta v^* \rangle = \langle v^* \rangle$ (Supplementary Note 3D). This is the hallmark of an adder.

### The full titration-switch model

To combine the activation switch with titration, we make the following assumptions (Supplementary Note 4A): 1) Since the affinities of ATP-DnaA and ADP-DnaA for their promoters are fairly similar[68], we assume that active and inactive DnaA have the same affinity for the promoter and only DnaA not bound to titration sites can repress the promoter; 2) the bound and free DnaA is activated and deactivated with the same rate; 3) Since the affinities of ADP-DnaA and ATP-DnaA for the titration sites are fairly similar[53], we assume that the affinities of inactive and active DnaA for the titration sites are the same; 4) replication is initiated when the free concentration of active DnaA reaches a threshold.

Using assumption 1, we model the change in the total number of DnaA proteins $N_T(t)$ in the cell via Eq. (1). To obtain the ATP-DnaA fraction, we model the change in the total number of ATP-DnaA proteins $N_D^{ATP}(t)$ explicitly. As newly produced DnaA proteins are more likely to bind ATP rather than ADP we add the DnaA production term to the change in the total number of ATP-DnaA proteins. Including again all known (de)activators as discussed in the LDDR model and using assumption 2, we obtain the following expression for the change in the number of ATP-DnaA proteins $N_D^{ATP}(t)$:

$$\begin{aligned} \frac{dN_D^{ATP}}{dt} = &\frac{\tilde{\phi}_p^0 \lambda V}{1 + \left( \frac{[D]_{T,f}}{K_D^p} \right)^n} \\ &+ \left( \alpha_l [l] V + \alpha_{d1} n_{ori}(t - \tau_{d1}) \right. \\ &\left. + \alpha_{d2}(t) n_{ori}(t - \tau_{d2}) \right) \frac{[D]_{ADP}}{K_D + [D]_{ADP}} \\ &- \left( \beta_{datA}(t) + \beta_{rida}(t) \right) n_{ori} \frac{[D]_{ATP}}{K_D + [D]_{ATP}}. \end{aligned} \qquad (9)$$

The active initiator concentration $[D]_{ATP}$ is obtained by dividing the number of ATP-DnaA proteins $N_D^{ATP}(t)$ by the volume $V(t)$ and the active initiator fraction $f(t)$ is obtained by dividing the number of ATP-DnaA proteins $N_D^{ATP}(t)$ by the total number of DnaA proteins $N_T(t)$. The total free DnaA concentration $[D]_{T,f}$ is obtained by using assumption 3) and solving a quadratic equation (Supplementary Note 2B2). Exploiting assumption 3 and fast binding and unbinding dynamics (Supplementary Note 2B2), the fraction $g = [D]_{ATP,f}/[D]_{T,f}$ of the concentration of free ATP-DnaA $[D]_{ATP,f}$ over the concentration of free total DnaA $[D]_{T,f}$ is equal to the fraction of the total ATP-DnaA concentration over the total DnaA concentration per cell $f = [D]_{ATP}/[D]_T$. The free ATP-DnaA concentration is therefore given by the concentration of free DnaA $[D]_{T,f}$ times the active fraction of DnaA $f$:

$$[D]_{ATP,f}(t) = [D]_{T,f}(t) \times f(t) \qquad (10)$$

### Reporting summary

Further information on research design is available in the Nature Research Reporting Summary linked to this article.

### Data availability

The datasets generated during and analyzed during the current study are available at Zenodo via https://doi.org/10.5281/zenodo.7057532.

## Code availability

The code is publicly available at the Github repository https://github.com/MareikeBerger/Cellcycleor at Zenodo via[114].

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

## Acknowledgements
We thank Lorenzo Olivi, Sander Tans, Suckjoon Jun, Erik van Nimwegen and Johan Elf for a careful reading of the manuscript. This work is part of the Dutch Research Council (NWO, grant number 024.003.019, P.R.t.W.) and was performed at the research institute AMOLF.

## Author contributions
M.B., and P.R.t.W. designed research; M.B. performed research; M.B. analyzed data; and M.B., and P.R.t.W. wrote the paper.

## Competing interests
The authors declare no competing interests.
