## [Peer Review File · Nature Communications]

Robust replication initiation from coupled homeostatic mechanismsREVIEWER COMMENTS

Reviewer #1 (Remarks to the Author):

The paper deals with the timely problem of developing a molecular-level model for the initiation of DNA replication in *E. coli*, a challenging, long-standing problem. Part of the difficulty of dealing with this problem is the large number of molecular processes involved (DARS, RIDA etc.) and it is unclear how they combine together to produce a stable cell cycle - which is the focus of the current paper. My main criticism is that while the authors put together different models that incorporate some of the ingredients, I could not see how the main text of the article supports the conclusions, and the strong assertion made by the authors that a combination of titration and switch models are *necessary* to make a robust cell cycle. The problem is that the landscape of models is quite vast, and there are endless tweaks that can be made to the models - and it is far from clear whether these conclusions are robust to such changes.

In general, the main text does not read well on its own and it feels that essentially all of the results are presented in the supplementary - the authors should make the main points of the paper supported by the main text.

Additional comments:

1. It was unclear why the switch models give an added.
2. It would be good to clarify why the titration model gives added correlations.
3. Does the negative autoregulation of the initiator have any effects on the initiation process?
4. Do the models occasionally give rise to an additional round of initiation in the cell cycle, and can the cells recover from such errors?

Reviewer #2 (Remarks to the Author):

This is very important and interesting study and is a most frontier study challenging total understanding of the replication regulation based on experimental results and mathematical formulation. The authors mathematically simulated the fluctuation of the ATP-DnaA level in dividing cells which contain some of known regulatory systems for DnaA. As quantitative regulatory systems for DnaA, the authors used: DnaA titration on the non-oriC loci on the chromosome (A). As activity regulatory systems, they used: DDAH (datA-dependent DnaA-ATP hydrolysis, D), RIDA (Regulatory inactivation of DnaA, R), and DARS1/2 (DnaA reactivation sequences, D), in addition to acidic phospholipid activation of DnaA (L), although the *in vivo* role for the acidic phospholipids in ATP/ADP-DnaA regulation remains still elusive lacking direct evidence. Also, the authors set various parameters of reaction rates in those systems. Virtual cells only with constant DnaA concentration and DnaA titration system (AIT model) show oscillation of the ATP-DnaA levels well only at a low growth rate ($T_d=2$ h). Virtual cells with the acidic lipid system, DDAH, DARS, RIDA in addition of constant DnaA concentration and DnaA titration system (LDDR switch model + titration model) show oscillation of the ATP-DnaA levels very well even at a high growth rate ($T_d=30$ min). In this process, virtual cells only with acidic lipid system and DDAH (LD switch model) or only with acidic phospholipid system, DDAH, DARS, and RIDA (LDDR switch model) were analyzed. Timely activation of DDAH, DARS, and RIDA is considered whereas the acidic lipid system is postulated to work constantly (which is referred later in this report). In addition, dnaA-expression regulator SeqA was included in the LDDR switch + titration model (Full model), showing its supportive role in the ATP-DnaA oscillation at higher growth rates. These comprehensive mathematical simulations are very important for understanding significance of the regulatory systems. I would like to suggest several points to discuss consistency of these simulations with *in vitro* mechanisms and *in vivo* phenotypes.

1. In these models, the acidic phospholipid system seems very important. However, as described in the supplemental material, the acidic phospholipid-null (pgsA-null mutant) cells are viable when a

specific mutation modulating membrane structure is present (eg. J Bacteriol. 2004 Oct;186(19):6526-35. doi: 10.1128/JB.186.19.6526-6535.2004; Microbiology (Reading). 2012 May;158(Pt 5):1238-1248. doi: 10.1099/mic.0.056945; at least one of these references should be cited). This fact should be included in the main text.

2. Related to the above. Although some papers suggest the role in ATP-DnaA reproduction by the acidic phospholipids, still direct evidence is lacking. (i) The sentences in the beginning part of page 8 should be revised not to mislead readers. (ii) Although Fug. S16G is shown, how is it possible to construct better simulation without the acidic phospholipid system ? This should be further analyzed. (ii) Even if the acidic phospholipids work for ATP-DnaA reproduction, it is reported to depend on oriC (J Biol Chem. 1992 Aug 25;267(24):16779-82.). This point is missed in the simulation. When ADP-DnaA bound to oriC is coincubated with the acidic phospholipids and ATP, it successfully becomes ATP-DnaA with the initiation activity. However, ADP-DnaA is simply coincubated with the acidic phospholipids, it becomes an inactive form in ATP binding, oriC binding, and the initiation, probably due to partial denaturation. At least, the oriC concentration should be considered as a function even in this system.

3. Overproduction of DnaA at several folds causes only slight stimulation of initiation and does not cause severe changes in the ATP-DnaA (%) in cells (in random cultures). Do the present simulations coincide with these facts?

4. Oversupply of oriC fragment by multicopy plasmid does not cause significant inhibition to replication initiation of the chromosomal oriC. Do the present simulations coincide with these facts? This might be related to the above question as changes in DnaA concentrations and effects of DnaA titration.

5. In all related data, showing the mean ATP-DnaA (%) in cells would be helpful in comparing with in vivo data.

6. For DARS2, a possible role for Fis is not included. Fis is required for activation of DARS2 and cellular expression of Fis varies at different growth conditions (eg. J Bacteriol. 2006 Aug;188(16):5775-82. doi: 10.1128/JB.00276-06.). This should be considered in the stimulations at low and high growth rates.

7. In the last part of Discussion, importance of the chromosome position of DARS2 is indicated. This is proved by the recent papers (PLoS Genet. 2016 Sep 2;12(9):e1006286. doi: 10.1371/journal.pgen.1006286; Genes Cells. 2016 Sep;21(9):1015-23. doi: 10.1111/gtc.12395.). These data should be considered in the present stimulations. Similarly, the position of *datA* can be taken into consideration based on experimental data.

8. The results of Full model including SeA should be included in the last part of Results of the main text.

Reviewer #3 (Remarks to the Author):

In this manuscript, the authors present a nice framework to combine two classes of models. This combination is a valuable attempt to bridge the gap between experimental observations of DnaA titration and DnaA-ATP/ADP switching. The authors also provide non-trivial predictions and proposed a new role of *dnaA* titration process to limit the variations. I believe that the models presented in the manuscript is valuable for understanding the replication initiation control mechanisms. Some points remain to be addressed.

1. The main point of this manuscript is focused on the robustness, defined by the CV of the initiation

mass, of the DNA replication initiation models. For the LD or LDDR model, the variation of initiation mass mainly comes from the noise of lipid concentration dynamics. However, for the titration model, the variation represents the effect of re-initiation (Fig. 5C). Can the authors justify why the variation of initiation mass are from two very different sources and compare to each other?

2. The authors first revised the classic initiator titration model (AIT model) that describes the regulation of replication initiation. They declared that this model is insufficient to explain experimental data because it produces re-initiation in fast-growth regime. To my knowledge, this phenomenon is parameter-dependent, e.g., number of DnaA boxes, distribution of DnaA boxes, DnaA synthesis rate, and etc. Also, it can be repressed by adding eclipse period. Although this issue is partly discussed in the combined model of LDDR + titration, it should also be discussed in the titration model to justify its weakness.

3. To establish the LD or LDDR model, different molecular processes responsible for DnaA-ATP/ADP conversion have been implied in the model. The rate for each process have been defined, with dependence on origin-density, replication fork number, copy number of DARS loci, and constant rate for DDAH, RIDA, DARS, and acidic phospholipids, respectively. While there are experimental evidences to support that DDAH and RIDA can convert DnaA-ATP into DnaA-ADP, DARS and acidic phospholipids acts contrarily, I wonder if there is any experimental evidence to directly support that the rate of these process quantitatively follow what have been defined in the model. In particular, given the fact that both DDAH and DARS are based on specific chromosomal loci, it is not obvious for me to expect the rate of DDAH to be origin-density-dependent while the activation rate of DARS is proportional to the number of DARS loci. In addition, the drop of β_{datA} shown in Fig. S7B is not obvious to me either.

4. In the LD model, the authors state that the DDAH, rather than the RIDA, play the key role of DnaA-ATP deactivation. This choice makes the model highly dependent of the *datA* location. However, Frimodt et al. have reported that the changing of the *datA* location around the chromosome, even located nearby the *ter* site, have no significant effect on the initiation timing and initiation mass (Doi:10.1371/journal.pgen.1006286).

Reply to Reviewers (Berger and Ten Wolde, *Robust replication initiation from coupled homeostatic mechanisms*)

We thank all three referees for their careful reading of the manuscript and their supportive and constructive comments and questions. Below we respond to these point by point. The changes made to the main text and the SI are highlighted in blue.

Response to reviewer #1

Comment: *“The paper deals with the timely problem of developing a molecular-level model for the initiation of DNA replication in E. coli, a challenging, long-standing problem.”*

Reply: We are pleased that the referee recognizes the importance of the problem.

Comment: *“My main criticism is that while the authors put together different models that incorporate some of the ingredients, I could not see how the main text of the article supports the conclusions, and the strong assertion made by the authors that a combination of titration and switch models are *necessary* to make a robust cell cycle. The problem is that the landscape of models is quite vast, and there are endless tweaks that can be made to the models - and it is far from clear whether these conclusions are robust to such changes.”*

Reply: We fully agree with the referee that the role of modelling is to make strong predictions that are not sensitive to the details of the model, and certainly not to details that are not supported by experiments. Our model shows that a titration-based mechanism must fail, inevitably, at higher growth rates (Fig. 2). The reason follows from a scaling argument. The DnaA synthesis rate scales with the growth rate, see Eq. 1. In contrast, the titration-site formation rate per origin is nearly independent of the growth rate: when the titration sites are homogeneously distributed, as experiments show (Refs. 28, 51), then the titration-site formation rate per origin is set by the DNA duplication rate, which indeed varies only little with the growth rate (Ref. 4). The protein synthesis rate thus increases faster with the growth rate than the titration-site formation rate, which means that at sufficiently high growth rates, i.e. when the cell division time τ_d becomes shorter than the time T_C to replicate the DNA (section S2B4), the mechanism fails to sequester DnaA proteins after replication initiation, leading to premature reinitiation (Fig. 2), and a dramatic rise in the variance of the initiation volume (Fig. 5). A titration-based mechanism could, however, work if the titration sites were located near the origin, but this is not what experiments show (Refs. 28, 51). Moreover, the number of titration sites and their affinity are constrained by the experimentally observed initiation volume and varying these parameters in concert cannot prevent the reinitiation events at high growth rates (see the response to referee 3). We therefore conclude that with a homogeneous distribution of titration sites, as observed experimentally, the titration mechanism will fail at sufficiently high growth rates; this mechanism is not sufficient for generating robust replication cycles at all growth rates. We emphasize that this prediction follows from a scaling of two timescales, which is insensitive to the details of the model. This prediction is robust.

Combining titration with the suppression of DnaA synthesis by SeqA during the ‘eclipse’ period of about 10 minutes following replication initiation can prevent premature reinitia-

tion events, but only at high growth rates: at intermediate growth rates the duration of the cell cycle becomes much longer than the eclipse period and premature reinitiation events will occur (Fig. 2), causing a strong rise in the variance in the initiation volume (Fig. 5). The activation switch is thus necessary to prevent premature reinitiation at all growth rates.

While a titration-based mechanism is not sufficient and the switch is essential, the converse is not necessarily true. Indeed, our model predicts that a switch-based mechanism could be sufficient for generating robust replication cycles at all growth rates. To answer directly the question of the referee, a titration-based mechanism is not essential.

This raises the question why the system has also evolved a titration-based mechanism. To address this, in the original manuscript we developed a model of the switch based on experimental data (LDDR model), and then added titration sites to this system. As Fig. 5 of our manuscript shows, the system that combines the switch with titration is more robust than the system based on the switch only. We thus concluded not only that titration is not sufficient and a switch essential, but also that titration can enhance the robustness of the switch.

The referee raises the question how generic this observation is. While the full switch model (LDDR) is based on a large body of experimental data, it does, as the referee points out, contain many parameters and hence it remains unclear how generic this prediction is. Prompted by the question of the referee, we therefore decided to approach this question differently. To show that a concentration cycle, as generated by titration, can generically enhance an activity cycle, as induced by the switch, we have now extensively studied a minimal switch model, the LD model. We optimized it subject to experimental constraints, such as the initiation volume and the maximum (de)activation rates. We then added to this optimal switch the titration mechanism, *keeping all the parameters of the switch (and also the titration system) the same*. As Fig. S14A shows, the combined system is more robust than the system based on the switch only. We then show mathematically in section S4A2 that a concentration cycle can generically enhance an activation cycle by increasing the “gain”, i.e. the steepness of the oscillations; the higher gain means that fluctuations in the cytoplasmic concentration of active DnaA propagate less to fluctuations in the initiation volume (see Fig. S14B). This idea is generic and does not depend on the details of the model. Titration can thus enhance the switch by making the system more robust to fluctuations in the switch components, increasing the precision of replication initiation.

Action: We have completely rewritten the section in the main text on combining titration with DnaA activation. We now emphasize more clearly than before that a system based on titration alone will fail inevitably, and is therefore not sufficient; a switch is essential. Conversely, we stress that a system based on a switch alone could be sufficient, but adding titration to the switch makes the system more robust to fluctuations in the switch components. We also mention that this effect follows from a generic mechanism, namely that adding a concentration cycle to an activation cycle can increase the gain. We refer to a new section in the SI, S4A2, and a new figure, Fig. S14, where we demonstrate this using a minimal model of the switch.

Comment: *“In general, the main text does not read well on its own and it feels that essentially all of the results are presented in the supplementary - the authors should make*

the main points of the paper supported by the main text.”

Reply and Action: We very much appreciate this point of the referee and motivated also by the comments of the other referees we have now moved material from the Supporting Information (SI) to the main text, and from the Discussion to the Results section. In particular, we have moved section S4A2 (of the submitted manuscript) on the titration-SeqA model from the SI to the main text, and moved the corresponding paragraph in the Discussion to the titration section (AIT model) in the Results; we have also integrated the corresponding Fig. S14 (of the submitted manuscript) into Fig. 2 of the main text. In addition, we have included Fig. S13J, on the Coefficient of Variation (CV) of the initiation volume in the full model that combines titration, SeqA, and, the switch, into Fig. 5 of the main text. We would like to thank the referee for this remark because it has improved the structure of the manuscript: In the Results section, we now first show why the titration mechanism must fail in the regime of overlapping replication forks (Fig. 2A/B). We then show that SeqA can rescue the titration mechanism but only at (very) high growth rates; at intermediate growth rates, premature reinitiation events still arise, which means that another mechanism is needed (Fig. 2C/D). This then naturally leads to the switch model, presented in the next section. Fig. 3A shows that a switch based on lipid-mediated activation and *datA*-mediated deactivation (LD model) generates stable rhythms, but that the amplitude of oscillations markedly decreases at higher growth rates; Fig. 3B shows that the full switch model (LDDR model) generates large amplitude oscillations at all growth rates. We then explain how adder correlations can arise from a size-sensing mechanism (Fig. 4), before we end by discussing the design logic of this system, i.e. why it combines a concentration cycle as generated by titration and SeqA with an activation cycle as driven by the switch. Fig. 5 shows that the CV of the initiation volume in a titration-based system rises dramatically at higher growth rates because of premature reinitiation; that SeqA can only reduce the CV at very high growth rates; and that the switch is thus essential. This figure also shows that titration can help the switch by increasing the precision of replication initiation. These are the central predictions from our work, and they are now all presented in the Results section of the main text. The SI mainly provides details on our mathematical models, and elucidates some of our predictions.

Comment: *“Additional comments:”*

“1. It was unclear why the switch model gives an adder.”

Reply: The components of the switch will fluctuate inevitably because of biochemical noise. These fluctuations will give rise to fluctuations in the initiation volume. If the switch components are neither degraded actively nor produced with strong feedback control, then the fluctuations in the switch components will relax to the mean on a timescale set by the growth rate, i.e. the cell division time. Because the mapping between the concentration and thus the activity of the switch components and the initiation volume is roughly linear and (de)activation of DnaA is faster than the growth rate, also perturbations in the initiation volume will regress on this timescale. If this timescale is indeed the cell-division time, a deviation of the initiation volume from its mean will, on average, be halved every cycle (see Fig. 4). This naturally gives rise to adder correlations, as can be shown mathematically rather straightforwardly (see Eqs. S49-S53 in section S3D).

Action: We have reworded the caption of Fig. 4 and the corresponding section of the main text, and refer to section S3D for the mathematical derivation.

“2. It would be good to clarify why the titration model gives adder correlations.”

Reply: We agree that the explanation was very elementary. The key feature of an adder is that the added volume is on average constant, independent of the last initiation volume (Refs. 8, 14, 15). In the AIT titration model, replication is initiated when the number of proteins that have been produced since the last initiation event equals the number of titration sites, irrespective of the last initiation volume. Since in the AIT model the proteins are produced at a rate that is proportional to the volume of the cell (Eq. S32), this number of accumulated proteins is proportional to the volume that has been added since the last initiation event. The added volume since the last initiation event is thus, on average, always the same, irrespective of the last initiation volume—the hallmark of an adder.

Action: We have added an explanation in the first paragraph of the section *“A stochastic model can recover the ... adder correlations ...”* on p.5 of the manuscript and we have rewritten our explanation in section S2B6.

“3. Does the negative autoregulation of the initiator have any effects on the initiation process?”

Reply: Our modelling predicts that negative autoregulation does not play a direct role in replication initiation. This is supported by recent experiments, which show that the average cell volume, initiation volume, and precision of replication initiation are only weakly affected in strains where the native *dnaA* promoter has been replaced by constitutive *dnaA* expression (Ref. 61). Following Hansen et al. (Ref. 26), we believe that negative autoregulation only plays an indirect role, by setting the growth-rate dependence of the total DnaA concentration. Experiments have revealed that the total DnaA concentration varies with the growth rate, anti-correlating with the initiation volume (Ref. 7). However, these experiments also show that the variation of both the initiation volume and the total DnaA concentration with the growth rate is weak, i.e. about 50% over a tenfold change of the growth rate. Since it is known that negative autoregulation tends to keep the total protein concentration independent of the growth rate (Refs. 80, 81), it therefore seems likely that negative autoregulation is crucial for constraining the growth-rate dependence of the total DnaA concentration and hence the initiation volume. How negative autoregulation, together with titration and the differential sensitivity of the promoter to DnaA-ATP and DnaA-ADP, sets the growth-rate dependence of the total DnaA concentration and the initiation volume, is beyond the scope of the current work.

Action: We have reworded the paragraph in the Discussion section on the dependence of the initiation volume and the total DnaA concentration on the growth rate, and how this is likely to be controlled by negative autoregulation.

“4. Do the models occasionally give rise to an additional round of initiation in the cell cycle, and can the cells recover from such errors?”

Reply: We thank the referee for this important question. We start by emphasizing that a distinction needs to be made between the behavior of the models in the presence versus absence of biochemical noise. In the absence of any biochemical noise, the switch-based models never exhibit premature reinitiation during the same cell cycle. In contrast, the model based on titration alone inevitably gives rise to premature reinitiation events in the same cell cycle in the regime of overlapping replication forks; this happens even when the model includes a SeqA-mediated refractory period of about 10 minutes following replication initiation during which replication cannot be reinitiated. This is one of the main points of our paper (see Fig. 2). The suppression of DnaA synthesis by SeqA following replication initiation can prevent premature reinitiation only at very high growth rates; at intermediate growth rates, a model based on titration and SeqA still gives rise to reinitiation events, even in the absence of any biochemical noise. The full model, including the switch, titration, and SeqA, does not exhibit reinitiation events in the same cell cycle, at any growth rate, in the absence of any biochemical noise. In the *presence* of biochemical noise, all models will exhibit premature reinitiation if they do not include the SeqA mediated refractory period. With a refractory period of about 10 minutes, as the experiments indicate (Refs. 52-54), the probability of premature reinitiation becomes negligible, except in the titration-only model, where it inevitably happens in the overlapping fork-regime, and even in the deterministic system, as mentioned above.

To address the second question: all systems are stable. When a premature reinitiation event happens, the cell does recover from these errors, in all models. Even the titration-based system is stable, i.e. the concentrations do not diverge. However, premature reinitiation events dramatically increase the coefficient of variation (CV) of the initiation volume, as Fig. 5 shows for the titration-based AIT model (the CV becomes even larger than observed experimentally (Refs. 3, 8), even when the system is deterministic).

Action: In the section on the AIT model, we now emphasize more strongly that premature reinitiation events will arise in a titration-based system, even when the refractory or so-called ‘eclipse’ period of SeqA is included. In the captions of Fig. 4 and Fig. 5 we mention that the models include an eclipse period of 10 minutes to prevent immediate reinitiation due to biochemical noise. In the section **Coupling titration with DnaA activation enhances robustness** we systematically address all the points made in the previous paragraph: We first stress that all our models are stable in the presence of noise (the concentrations do not diverge), but that the precision of replication initiation varies markedly between the different models. We explain that a system based on titration alone will generate premature reinitiation at sufficiently high growth rates, causing the CV of the initiation volume to rise dramatically, even in a deterministic system. We then mention that the transient suppression of DnaA synthesis after replication initiation by SeqA can prevent premature reinitiation but only at high growth rates; at intermediate growth rates the switch is essential for preventing premature reinitiation.

Response to reviewer #2

Comment: *“This is a very important and interesting study and is a most frontier study challenging total understanding of the replication regulation based on experimental results and mathematical formulation.”*

Reply: We are pleased with the positive judgment of the referee.

Comment 1: *“In these models, the acidic phospholipid system seems very important. However, as described in the supplemental material, the acidic phospholipid-null (*pgsA*-null mutant) cells are viable when a specific mutation modulating membrane structure is present (eg. *J Bacteriol.* 2004 Oct;186(19):6526-35. doi: 10.1128/JB.186.19.6526-6535.2004; *Microbiology (Reading).* 2012 May;158(Pt 5):1238-1248. doi: 10.1099/mic.0.056945; at least one of these references should be cited). This fact should be included in the main text.”*

Reply and Action: We thank the referee for pointing out these papers. We now cite the first in the main text and both in the SI. As we describe under the next point, we also analyze these observations in the Discussion section of the main text and then in more detail in a new section S5A of the SI.

Comment 2i: *“Although some papers suggest the role in ATP-DnaA reproduction by the acidic phospholipids, still direct evidence is lacking. (i) The sentences in the beginning part of page 8 should be revised not to mislead readers.”*

Reply and Action We fully agree with the referee that the role of the lipids in rejuvenating DnaA remains poorly understood, as we discuss in more detail below. We have therefore rewritten this paragraph. Moreover, we have added a new paragraph to the Discussion of the main text and a whole new section to the SI, section S5, in which we analyze the role of the lipids in much more detail, as described under the next two points.

Comment 2ii: *“(ii) Although Fig. S16G is shown, how is it possible to construct better simulation without the acidic phospholipid system? This should be further analyzed.”*

Reply We thank the referee for this pertinent question. Indeed, while the in vitro experiments of Refs. 43, 55, 56 and in vivo experiments of Refs. 48, 57 support the idea of our model that acidic phospholipids promote the exchange of DnaA-bound ADP for ATP, Shiba et al. observed that the lethal effect of a *pgsA* null mutation, which causes a complete lack of the major acidic phospholipids, is alleviated by mutations that change the membrane structure (Ref. 78, SI Refs. 68, 69). Moreover, Camsund et al. reported that while downregulating *pgsA* reduced the growth rate, the initiation volume was not significantly altered (Ref. 79). These experiments hint that the lipids are not vital to the rejuvenation of DnaA. Prompted by these observations and the question of the referee, we have therefore analyzed scenarios in which lipid-mediated DnaA activation is absent. Our modelling predicts that lipid-mediated DnaA activation is essential for the switch, both in the LD and in the LDDR model (Fig. S20A-D). The capacity of the switch to act as an origin-density sensor hinges on the idea that the activation and deactivation rates scale differently with the origin density. The lipid-devoid LD model has only one

activation mechanism, protein synthesis, and one deactivation mechanism, *datA*. Since the rate of protein synthesis is independent of the origin density, while that of *datA* scales with the origin density, the system is stable; however, to keep the initiation volume in the experimentally observed range, the deactivation rate must be smaller than or comparable to the growth rate, yielding oscillations of weak amplitude. The LDDR model has, besides the lipids, protein synthesis and *datA*, also *DARS1/2* and RIDA as an activation and deactivation mechanism, respectively. When the lipids are taken out, only protein synthesis remains as a (de)activation mechanism that does not scale with the origin density. As a result, to keep the system stable (see Fig. S9), the rates of all other activation and deactivation mechanisms must be comparable to or lower than the growth rate, which sets the protein synthesis rate. This, again, dramatically reduces the amplitude of the oscillations (Fig. S20A-D), to such a degree that they likely do not persist in the presence of the inevitable biochemical noise. The full model, which also includes titration and SeqA, is, however, surprisingly robust to the removal of the lipids (Fig. S20E-G). This is interesting, because while a system based on only a lipid-devoid switch merely produces very weak oscillations (Fig. S20A-D) and a system based on only titration and SeqA still exhibits premature re-initiation events at intermediate growth rates (Fig. 2C), the combined system generates robust oscillations over the full range of growth rates (Fig. S20G). In the full model, the switch rescues the titration-SeqA mechanism by preventing reinitiation, while, conversely, titration helps the unstable lipid-devoid switch by making it stable. Nonetheless, while the lipid-devoid system is remarkably robust, it is less robust than the full model with lipid-mediated DnaA activation (Fig. S20G).

Action: We have included a new section in the SI, section S5, in which we analyze the role of the lipids in detail. In section S5A, we first describe the experiments that support the idea that lipids promote the exchange of ADP for ATP, and then we describe the experiments by Shiba et al. and Camsund et al. We then analyze the LD, LDDR, and the full model, with lipid-mediated DnaA activation taken out entirely. In the main text, we discuss the experiments supporting and questioning the role of the lipids in the Discussion section, where we also summarize the principal findings of our analysis of the models without the lipids. In section S5B, we then analyze the role of *oriC* in rejuvenating DnaA, as discussed under the next point.

Comment 2iii: *“(iii) Even if the acidic phospholipids work for ATP-DnaA reproduction, it is reported to depend on oriC (J Biol Chem. 1992 Aug 25;267(24):16779-82.). This point is missed in the simulation. When ADP-DnaA bound to oriC is coincubated with the acidic phospholipids and ATP, it successfully becomes ATP-DnaA with the initiation activity. However, ADP-DnaA is simply coincubated with the acidic phospholipids, it becomes an inactive form in ATP binding, oriC binding, and the initiation, probably due to partial denaturation. At least, the oriC concentration should be considered as a function even in this system.”*

Reply: This is a truly excellent question. The picture that emerges from these experiments is confusing. Below, we first list the relevant experimental observations. We then discuss what we conclude from these observations, and which possible lipid-mediated DnaA reactivation scenarios would be consistent with the data. We end by describing how we have studied these scenarios in our simulations.

The experimental observations are: **1)** DnaA can bind ATP and ADP in the absence of phospholipids, see Figs. 1, 2, 4 and 6 of Ref. 43. DnaA bound to oriC can also bind ATP, see Ref. 55. **2)** Acidic phospholipids such as cardiolipin (CL) and phosphatidylglycerol (PG) enhance the release of ADP and ATP from DnaA, see Figs. 1 and 2 of Ref. 43, Figs. 1 and 4 of Yung and Kornberg (PNAS, 1988), and Fig. 3 of Ref. 55. **3)** Phospholipids can restore replication activity of DnaA bound to ADP, see Fig. 3 of Ref. 43 and Fig. 2 of Yung and Kornberg (PNAS, 1988). **4)** CL blocks the binding of ATP to DnaA, see Fig. 6 of Ref. 43. **5)** CL can inactivate nucleotide-free DnaA for replication initiation, see Fig. 7 of Ref. 43. **6)** Incubating DnaA-ADP or DnaA-ATP in the presence of phospholipids inhibits the binding of DnaA-ADP/ATP to oriC, see Fig. 1 of Ref. 55. **7)** Phospholipids speed up the dissociation of DnaA-ATP and DnaA-ADP from oriC but only marginally, see Fig. 2 of Ref. 55. **8)** (a) Incubating DnaA-ADP with the lipids and the oriC restores replication activity (irrespective of whether the other replication components were added initially or later), see 2nd and 3rd row Table I of Ref. 55. (b) but first incubating DnaA-ADP with the lipids and then adding oriC later does not, see 4th row of the same table. The temporal order in which the lipids and oriC are added to DnaA-ADP thus seems to matter (but see below). **9)** Lipids lower the *apparent* affinity between DnaA and ATP, yet the presence of oriC during the incubation of DnaA with phospholipids restores the affinity between DnaA and ATP, see Table 2 of Ref. 55. **10)** Binding of DnaA-ADP and DnaA-ATP to oriC slows down the release of the nucleotides as stimulated by the lipids, see Figs. 3 and 4 of Ref. 55.

Emerging picture The picture that emerges from these studies is that acidic phospholipids and nucleotides mutually exclude each other in binding DnaA. This could explain the observation that lipids both promote the release (point 2) and inhibit the binding (point 4) of nucleotides; in turn, the release-promoting effect could explain why lipids can restore replication-initiation activity (points 3 and 8(a)). In addition, the lipids also compete with oriC for binding DnaA. This may explain why adding lipids impedes oriC binding (point 6) and hence replication initiation (point 5 and point 8(b)) and why it stimulates the dissociation of DnaA-ADP/ATP from oriC (point 7; the fact that this effect is weak indicates that the complex of oriC and DnaA is very stable). At the same time, DnaA retains its high affinity for nucleotides when bound to oriC (point 9). These observations are consistent with structural studies (see Saxena review, Ref. 49), which show that nucleotides and DNA (oriC) bind to different domains of DnaA, while the lipids bind to the domain (III) that also binds the nucleotides (hence the mutually exclusive binding) yet on the border with the domain (IV) that binds oriC (which could explain why oriC and lipid binding are mutually exclusive).

Open questions The biggest open question is how to reconcile the observation that lipids in a mixture of DnaA-ADP, oriC, ATP and other replication components can initiate DNA synthesis (point 8(a)), while ADP is much less likely to be released from DnaA by the lipids when the DnaA-ADP is bound to oriC (point 10). The observation of Crooke et al. (Ref. 55) that the temporal order in which oriC and lipids are added to DnaA-ADP (point 8) is particularly confusing, because the reactions involved are association-dissociation reactions (i.e. DnaA-lipid binding, DnaA-oriC binding, and DnaA-(oriC)-nucleotide binding), which can reach thermodynamic equilibrium; and in the equilibrium state, the temporal order in which the components have been added should be irrelevant. How could we interpret

these results? It could be that the reactions do not reach equilibrium, in which case it becomes very hard to draw a firm conclusion. It is also conceivable that the (relative) concentrations of the components matter, because reactions with competitive binding, as seems to be the case here, can lead to non-trivial dependencies of the bound fractions on the relative concentrations. This is particularly pressing because the concentrations in the in vitro experiments are likely to be very different from the effective concentrations in vivo; the latter depend not only on the concentrations of the lipids in the membrane, but also on the membrane area and the cytoplasmic volume of the cell.

To make progress, we considered three different scenarios:

1. DnaA-ADP generated by DDAH (via *datA*) and RIDA binds the lipids, causing the ADP to dissociate. After DnaA then dissociates from the lipids, it rapidly binds ATP (point 1 above). DnaA-ATP can then bind *oriC*.
2. DnaA-ADP binds to *oriC*, and then DnaA-ADP-*oriC* interacts with the lipids, leading to the exchange of ADP for ATP.
3. DnaA-ADP binds the lipids, ADP is released, *oriC* moves to the DnaA bound to the lipids, causing ATP to bind.

The first scenario is that considered in our manuscript. It is clear that experiments support the idea that DnaA-ADP can bind the lipids, leading to the release of ADP, that DnaA can bind ATP in the cytoplasm, and that DnaA-ATP can bind *oriC*. The only open question is on what timescale DnaA dissociates from the membrane. The second scenario is consistent with observation 8(a). However, as discussed above, it seems at odds with the observation that lipids are less likely to prompt the release of ADP when the DnaA-ADP is bound to *oriC* (point 10). Moreover, there is no evidence that *oriC* is associated with the membrane (Ref. 61). The same criticism applies to scenario 3.

While scenarios 2 and 3 seem less likely in our opinion, they can also not be ruled out. As the referee alludes to, in these scenarios the rejuvenation of DnaA is contingent on *oriC*, making the activation reaction dependent on the origin density. We have therefore also developed models in which the lipid-mediated DnaA activation is proportional to the origin density. Given the ambiguities in the experimental observations as discussed above, and the lack of quantitative, time-series data like that obtained for the Kai system (Refs. 99 and 100 of SI), developing a detailed mathematical model, which includes the competitive DnaA binding between the lipids, nucleotides, and *oriC*, is at this stage not feasible. We have therefore developed coarse-grained models similar to those of the main text (see section S5B). We find that the robustness of the switch is severely compromised when the lipid-mediated activation rate is proportional to the origin density. The reason is that the *datA*-mediated DnaA deactivation rate and the lipid-mediated DnaA activation rate now scale with the origin density in the same manner, which means that the lipid-mediated activation reaction merely reduces the deactivating effect of *datA*, lowering the amplitude of the oscillations. Effectively, the models become very similar to those in which the lipids are taken out entirely (section S5A), because protein synthesis is the only (de)activation mechanism that does not scale with the origin density. Indeed, the LD and LDDR models with an origin-dependent lipid-mediated activation rate behave very poorly (Fig. S20B-D), generating only weak oscillations, while the full model (titration-SeqA-switch) with an

origin-dependent lipid-mediated DnaA activation rate is again surprisingly robust (Fig. S20E/F).

Action: We have added a new section to the SI, S5B, where we first discuss the experimental observations, then the picture that emerges from these observations, the open questions, and the scenarios that could be envisioned. We then analyze in section S5B2, the LD, LDDR, and full model with a lipid-mediated DnaA activation rate that scales with the origin density, explaining why these models behave very similarly to the respective models in which the lipids are taken out entirely. We also briefly describe the principal finding in the Discussion section of the main text.

Comments 3: *“Overproduction of DnaA at several folds causes only slight stimulation of initiation and does not cause severe changes in the ATP-DnaA (%) in cells (in random cultures). Do the present simulations coincide with these facts?”*

Reply: There are two questions. To answer the second question first: our model indeed predicts that the fraction of active DnaA does not change significantly if the total DnaA concentration is varied (see Fig. S10), provided that the (de) activation rates are higher than the growth rate; the latter is, in fact, what the measured deactivation rates of *datA* and RIDA show, and which also must be true generically for generating high-amplitude oscillations. The first question is how the initiation volume changes when DnaA is overproduced. Experimentally, the total DnaA concentration in *E. coli* cells can be varied by introducing plasmids containing inducible *dnaA* promoters into cells (Refs. 76, 77), by replacing the native *dnaA* promoter by an inducible promoter (Ref. 61) or by repressing the native *dnaA* promoter further using tunable CRISPR interference (Ref. 4). All of these experiments reported a negative dependence of the initiation volume on the total DnaA concentration (Refs. 4, 61, 76, 77). Furthermore, Zheng et al. measured the average DnaA concentration and the initiation volume at different growth rates and also reported a negative correlation between these two variables (Ref. 7). While Flatten et al. (Ref. 56 of SI) reported that upon increasing the total DnaA concentration two-fold the average volume per number of origins decreased only very weakly, they also observed a decrease in the volume per number of origins when the total DnaA concentration was increased even further (up to 35 times the wild type concentration) (Ref. 53 of SI). To test the effect of varying the total DnaA concentration in our full model we mimic the plasmid experiments of Hill et al. (Ref. 76) and Atlung and Hansen (Ref. 77) by adding an external production term of DnaA to our full Switch-titration-SeqA model. As observed experimentally, in the full model the initiation volume decreases with increasing DnaA total concentration for a wide range of growth rates (Fig. S17). More specifically, we find that the initiation volume per origin is approximately inversely proportional to the total DnaA concentration, such that for a doubling of the DnaA concentration the initiation volume is approximately halved. This agrees quantitatively with the finding of Zheng et al. who observe that for a decrease of the total concentration of about 20%, the average initiation mass per number of origins increases by about 20% (see Extended Data Fig. 5 of Ref. 7). Also Atlung and Hansen (Ref. 77) find an almost linear increase in the average number of origins per mass with increasing total DnaA concentration up to at least two times the wild type DnaA total concentration (corresponding to an inverse relationship between the average volume per origin and the total DnaA concentration). Therefore, our simulations are indeed in

good quantitative agreement with the experimental findings of varying the total DnaA concentration.

Action: We have added a new section S4B3 to the SI, in which we discuss the experiments in which DnaA is overproduced. We have also included a new Figure, Fig. S17. Moreover, we briefly discuss these results in the Discussion section of the main text (in paragraph on model validation), referring to section S4B3 for more details.

Comment 4: *“Oversupply of oriC fragment by multicopy plasmid does not cause significant inhibition to replication initiation of the chromosomal oriC. Do the present simulations coincide with these facts? This might be related to the above question as changes in DnaA concentrations and effects of DnaA titration.”*

Reply: The oriC region and hence the multicopy plasmids do not affect the switch, because the oriC region is neither an activator nor a deactivator of DnaA. The origin region does however contain several high and low affinity binding sites for DnaA and as such could affect the titration mechanism. Indeed, a study by Christensen et al. (Journal of Bacteriology, 1999) shows that introducing different multicopy plasmids, carrying different numbers of DnaA binding sites, into *E. coli*, leads to different changes in the initiation volume per origin. They observed that for all growth rates studied, plasmids with a higher number of DnaA boxes cause a larger decrease in the chromosomal origin concentration and hence a larger increase in the initiation volume per origin. This finding is consistent with the predictions from our full model: The initiation volume increases with the increasing number of titration sites per origin (Fig. S15). Interestingly, the agreement is not only qualitative but even nearly quantitative, without any additional fitting. Specifically, the degree to which the initiation volume changes depends on the relative change in the number of titration sites per origin and the growth rate. Focusing on the strain carrying plasmid pFHC946, which contained all of the DnaA boxes from the *oriC* region: this showed about a 10% increase in the initiation volume per origin at a high growth rate, and a nearly 20% change in the initiation volume per origin at a low growth rate (Fig. 6 of Christensen paper). To compare this observation to the predictions from our model, we note that at the high growth rate, the number of pFHC946 plasmids per origin is about 13 while at the low growth rate it is about 57 (Table 1, Christensen paper). To estimate the total number of DnaA titration sites that these plasmid copies carry, we consider that plasmid pFHC946, like the chromosomal oriC region, contains two high affinity DnaA binding sites (R1 and R4) and one intermediate affinity site R2, resulting in 2-3 medium-high affinity titration sites per plasmid copy. The total number of titration sites on the plasmid copies combined is, per chromosomal origin, thus $13 \times (2 - 3) \approx 25 - 40$ at high growth rates and $57 \times (2 - 3) \approx 100 - 150$ at low growth rates. Our model predicts that at high growth rates (Fig. S15D, red line), a change in the number of titration sites from $n_s = 300$, corresponding to wild-type cells with no extra plasmids, to $n_s \approx 350$, corresponding to the pFHC946 strain, causes a relative change in the initiation volume of 10%, in quantitative agreement with the experiments of Christensen et al. At low growth rates, our model predicts (Fig. S15A, red line) that a change in the number of titration sites, from 300 in wild-type cells to 400 – 450 in the cells of the pFHC946 strain, causes a change in the initiation volume of about 22 – 35%, in near quantitative agreement with the reported change of about 20%. We believe this near quantitative agreement with experiments is remarkable, since this was

a blind prediction, with no fitting involved. We therefore would like to thank the referee for prompting us to make this comparison.

Action: We have added a new subsection S4B2 to the SI, where we describe this comparison. Moreover, we briefly discuss this in the Discussion section of the main text (in paragraph on model validation), referring to section S4B2 for more details.

Comment 5: *“In all related data, showing the mean ATP-DnaA (%) in cells would be helpful in comparing with in vivo data.”*

Reply and Action: This is a useful suggestion, which we have incorporated.

Comment 6: *“For DARS2, a possible role for Fis is not included. Fis is required for activation of DARS2 and cellular expression of Fis varies at different growth conditions (eg. J Bacteriol. 2006 Aug;188(16):5775-82. doi: 10.1128/JB.00276-06.). This should be considered in the simulations at low and high growth rates.”*

Reply: The activity of *DARS2* is indeed not only modulated by IHF but also by Fis, and their activities may not only be temporally regulated but also depend on the growth rate. Experiments indicate that IHF temporally regulates the activity of *DARS2* over the course of the cell cycle, which we modelled via a step function with a high-activity state α_{d2}^+ and a low-activity state α_{d2}^- , see Fig. S7. There is, to our knowledge, no clear data on the temporal regulation of Fis over the course of the cell cycle, and as we wrote in section S3A we therefore absorbed the contribution of Fis into the values of α_{d2}^+ , α_{d2}^- . We agree with the referee that there is evidence that the activity of Fis increases with the growth rate. However, as we wrote in section S3A, *“precisely how the binding of Fis depends on the growth rate of the cell remains to be determined. Since α_{d2}^+ contributes to the initiation volume only in the high-growth rate regime of overlapping replication forks, while α_{d2}^- only (weakly) contributes to the initiation volume at low growth rates, . . . , we assume, for simplicity, that the values of α_{d2}^+ and α_{d2}^- are independent of the growth rate.”* We did realize, however, that a constant activity independent of the growth rate could negatively affect robustness at low growth rates. As we wrote in section S3C2, *“A short time τ_{d2} [after replication] . . . the activity of DARS2 increases and the active fraction rises. In this low growth rate regime, the active fraction is therefore high for a large fraction of the cell cycle . . . The robustness of the LDDR model at low growth rates could be enhanced by reducing the activation rate of DARS2 specifically at low growth rates, because this would then in this regime lead to a slower rise in the active fraction when DARS2 is duplicated. Indeed, only in the high growth rate regime, DARS2 is essential to vigorously counteract the strong deactivator RIDA, enabling a new round of replication while the old round has not finished yet. There is experimental evidence that the activity of DARS2 decreases with decreasing growth rate of the cell, supporting this idea [11]. Importantly however, titration naturally enhances the robustness of the switch in the low growth rate regime, by sharpening the oscillations in the concentration of free, active DnaA”*.

Prompted by the comment of the referee, we now explicitly show in Fig. S8C that a lower activity of *DARS2* at lower growth rates positively affects the shape of the oscillations. However, we iterate that titration already enhances the shape of the oscillations. Moreover,

the functional dependence of *DARS2* activity on the growth rate, as determined by Fis (and IHF), has not been characterized. Most importantly, we find that *even with the constant, high activity, DARS2* has only a weak impact on the initiation volume at low growth rates, see Fig. S16I. We have therefore decided not to change our baseline parameter values.

Action: We now show oscillations in the LDDR model with a lower *DARS2* activity at a low growth rate (Fig. S8C), and we have slightly reworded the second paragraph of section S3C2 to discuss this point more clearly. We also refer to our model validation section S4B and Fig. S16 for a more detailed comparison against experimental data.

Comment 7: “*In the last part of Discussion, importance of the chromosome position of DARS2 is indicated. This is proved by the recent papers (PLoS Genet. 2016 Sep 2;12(9):e1006286. doi: 10.1371/journal.pgen.1006286; Genes Cells. 2016 Sep;21(9):1015-23. doi: 10.1111/gtc.12395.). These data should be considered in the present stimulations. Similarly, the position of datA can be taken into consideration based on experimental data.*”

Reply: We thank the referee for pointing out these very relevant papers. Interestingly, we had, in fact, already performed simulations in which the locus of *DARS2* was moved along the chromosome (see Fig. S16 of previous and current SI). We predicted that translocating *DARS2* to the terminus increases the initiation volume, especially at high growth rates (Fig. S16I). Remarkably, this is precisely what Inoue *et al.* (Genes to Cells, 2016) found. Motivated by the suggestion of the referee, and the papers he/she points out, we have now also studied the effect of translocating *datA*. The experiments reveal that moving *datA* towards the terminus can have two effects: 1) change the initiation volume and 2) cause premature reinitiation. Concerning the first observation, it has been observed that placing *datA* near the terminus decreases the initiation volume per origin v^* (Frimodt-Møller, PLoS Genetics, 2016). Also our model predicts that at high growth rates the initiation volume decreases when *datA* is moved towards the terminus, see Fig. S16J. This finding can be understood by noting that translocating *datA* to the terminus lowers its effective copy number and thereby reduces the effective deactivation rate, leading to a lower initiation volume per origin. This effect becomes stronger at higher growth rates, where the cell-doubling time becomes shorter compared to the time to replicate the chromosome. In contrast, at lower growth rates, it takes longer before replication is initiated during the cell cycle, such that the effect of moving the position on the effective copy number will be smaller. Indeed, our model predicts that at low growth rates the initiation volume does not decrease; in fact, it increases weakly (Fig. S16J). This is because of a second, *spatio-temporal* effect (rather than a change in the *average* copy number): moving *datA* towards the terminus means that *datA* will be doubled *later* in the cell cycle and therefore also the activity of *datA* will increase later in the cell cycle. This means that, if a new round of replication has not yet been initiated, the active DnaA concentration will reach the initiation threshold later, *increasing* the initiation volume. There are thus two competing effects, and which one dominates depends on the growth rate. To our knowledge, it has not been measured how the initiation volume changes when *datA* is translocated towards the terminus at low growth rates; we thus regard this as a novel prediction from our model. Concerning the second observation, the premature reinitiation events: experiments have shown that moving *datA* towards the terminus leads to premature reinitiation at high (Kitagawa *et al.*, Genes and Development, 1998)), but not at low growth rates (Frimodt-Møller,

PLoS Genetics, 2016; Kitagawa et al., Genes and Development, 1998). Our model can qualitatively reproduce these observations (see Fig. S16J). According to our model, these premature reinitiations are related to the second spatio-temporal effect, which perturbs the shape of the temporal oscillations as generated by *datA*, *DARS2* and RIDA: the active fraction first decreases after initiation due to a high activity via RIDA, then rises due an increase in *DARS2* activity and then decreases *again* due to a high *datA* activity before the active free concentration rises and reaches the initiation threshold. At higher growth rates, these “double” oscillations can lead to premature reinitiation events (Fig. S16J, shaded area). In contrast, at low growth rates, the effect of titration on shaping the oscillations of the active free concentration is much stronger, and this can protect the system from premature reinitiation (Fig. S16J, blue line). The effect of moving *datA* can thus be highly non-trivial, and while we fully realize how difficult these experiments would be, it would reveal a wealth of information if time traces of active DnaA could be obtained experimentally.

Action: We added the prediction on moving the *datA* locus to section S4B, and we have added a corresponding panel J to Fig. S16, in which we show the effect of moving the *datA* locus on the initiation volume. Moreover, we now compare our predictions with the data of Frimodt-Møller et al. and Inoue et al., both on moving the locus of *DARS2* and that of *datA*. Furthermore, we now comment on these observations in the Discussion section of the main text, where we refer for details to the SI section S4B.

Comment 8: “*The results of Full model including SeqA should be included in the last part of Results of the main text.*”

Reply and Action: We thank the referee for this suggestion, which we have embraced. We have moved section S4A2 (of the submitted SI) on the titration-SeqA model from the SI to the main text, and moved the corresponding paragraph in the Discussion to the titration section (AIT model) in the Results; we have also integrated the corresponding Fig. S14 (of the submitted SI) into Fig. 2 of the main text. In addition, we have included Fig. S13J, on the Coefficient of Variation (CV) of the initiation volume in the full model that combines titration, SeqA, and, the switch, into Fig. 5 of the main text. We believe these changes have significantly improved the structure of the main text.

Response to reviewer #3

Comment: “*In this manuscript, the authors present a nice framework to combine two classes of models. This combination is a valuable attempt to bridge the gap between experimental observations of DnaA titration and DnaA-ATP/ADP switching. The authors also provide non-trivial predictions and proposed a new role of dnaA titration process to limit the variations. I believe that the models presented in the manuscript is valuable for understanding the replication initiation control mechanisms.*”

Reply: We thank the referee for his / her kind words.

Comment 1: “*The main point of this manuscript is focused on the robustness, defined by the CV of the initiation mass, of the DNA replication initiation models. For the LD or LDDR model, the variation of initiation mass mainly comes from the noise of lipid*

concentration dynamics. However, for the titration model, the variation represents the effect of re-initiation (Fig. 5C). Can the authors justify why the variation of initiation mass are from two very different sources and compare to each other?”

Reply: While the switch yields a non-zero variance in the initiation volume only in the presence of biochemical noise, the titration mechanism yields a non-zero variance in the initiation volume even in the absence of any biochemical noise. We believe the latter is one of the most important, non-trivial, findings of our study, and, indeed, for this reason we decided to show the results for the titration mechanism in the absence of biochemical noise. Adding biochemical noise would simply increase the variance in the initiation volume.

Yet, we understand the question of the referee. In Fig. S13K we therefore included the same source of noise in the DnaA concentration in all models. The results are qualitatively very similar to Fig. 5C in the main text. This is because the underlying mechanisms are the same: A system that is only based on titration will fail at high growth rates because of premature reinitiation events, and SeqA can only prevent these at very high growth rates. To generate robust rhythms at all growth rates, a switch is essential. Conversely, titration can help the switch by sharpening the oscillations, such that fluctuations in the switch propagate less to the initiation volume, as we now not only show in Fig. 5C, but also explain mathematically in a new section S4A2.

Action: We have rewritten the section **Combining titration with DnaA activation enhances robustness**. We now emphasize that the premature reinitiation events in the AIT titration model cause the Coefficient of Variation (CV) of the initiation volume to dramatically rise with the growth rate, even in a deterministic system with no noise. We then point out that SeqA can prevent these only at high growth rates, and that the switch is essential to generate robust rhythms at all growth rates. We then explain intuitively how titration can enhance the switch, and we refer to the new section S4A2 for the mathematical derivation. In the caption of Fig. 5 we point out that the same trends are observed in the models with noise in DnaA production, referring to Fig. S13.

Comment 2: *“The authors first revised the classic initiator titration model (AIT model) that describes the regulation of replication initiation. They declared that this model is insufficient to explain experimental data because it produces re-initiation in fast-growth regime. To my knowledge, this phenomenon is parameter-dependent, e.g., number of DnaA boxes, distribution of DnaA boxes, DnaA synthesis rate, and etc. Also, it can be repressed by adding eclipse period. Although this issue is partly discussed in the combined model of LDDR + titration, it should also be discussed in the titration model to justify its weakness.”*

Reply: We thank the referee for this important point. The titration mechanism gives rise to reinitiation events at high growth rates because new proteins are produced faster than new titration sites. As we show in Fig. S3, placing all titration sites close to the origin can suppress these reinitiation events: Upon replication initiation, all titration sites are doubled nearly instantly and the free DnaA concentration drops rapidly, thus preventing reinitiation at high growth rates (Fig. S3B). Experiments report however that the titration sites are distributed randomly on the chromosome (Ref. 28, 51), giving rise to reinitiation events at high growth rates (Figs. 2B and S3D). We therefore decided

to present the scenario of a homogeneous titration site distribution in the main text in Fig. 2.

The premature reinitiation events can be prevented by the transient suppression of DnaA synthesis by SeqA after replication initiation during the ‘eclipse’ period, but only at high growth rates, as we now show in Fig. 2D (Fig. S14 in originally submitted SI). At intermediate rates, the duration of the cell cycle becomes however significantly longer than this eclipse period, such that this protective effect becomes much weaker, and premature reinitiation events arise (Fig. 2C).

The referee also raises the question how the failure of the titration mechanism depends on the number of titration sites and the DnaA synthesis rate. The DnaA synthesis rate is constrained by the observations that DnaA is not actively degraded and the total concentration is only weakly dependent on the growth rate; it is thus set by the growth rate. Indeed, the titration mechanism fails precisely because the DnaA synthesis rate scales faster with the growth rate than the titration-site formation rate does: the protein synthesis rate scales with the growth rate (see Eq. 1), while the titration-site formation rate per origin, which is set by the DNA duplication rate, is nearly independent of the growth rate (section S2B4). The two remaining parameters in the titration model are the number of titration sites and their binding affinity. But these are constrained by the initiation volume: to keep the initiation volume constant, the binding affinity must decrease as the number of titration sites is increased. Importantly, also when these parameters are varied (jointly, to keep the initiation volume constant), the titration-based mechanism still gives rise to premature reinitiation events at high growth rates (see Fig. S3E). Because the failure of the titration mechanism arises from the different scaling of two timescales, this prediction of our model is robust, i.e. insensitive to the precise details of our model.

Action: Following the suggestion of the referee, we now discuss the role of SeqA in the titration section. Moreover, we have integrated Fig. S14 from the (original) SI into Fig. 2 of the main text. In the main text, we thus first present the failure of the titration mechanism at intermediate and high growth rates, and describe how this depends on the number of titration sites, their affinity, and their spatial distribution, referring to an expanded section S2 for details, showing the results on the number of titration sites and their affinity in Fig. S3E. We then explain that the titration-mechanism can be rescued by SeqA, but only at high growth rates, not at intermediate growth rates. This then leads us to conclude that another mechanism is needed, which is presented in the next section. We thank the referee for her / his suggestion to move the discussion on SeqA to the titration section, because it has significantly improved the structure of the paper.

Comment 3: *“To establish the LD or LDDR model, different molecular processes responsible for DnaA-ATP/ADP conversion have been implied in the model. The rate for each process have been defined, with dependence on origin-density, replication fork number, copy number of DARS loci, and constant rate for DDAH, RIDA, DARS, and acidic phospholipids, respectively. While there are experimental evidences to support that DDAH and RIDA can convert DnaA-ATP into DnaA-ADP, DARS and acidic phospholipids acts contrarily, I wonder if there is any experimental evidence to directly support that the rate of these process quantitatively follow what have been defined in the model. In particular, given the fact that both DDAH and DARS are based on specific chromosomal loci, it is not obvious for me to*

expect the rate of DDAH to be origin-density-dependent while the activation rate of DARS is proportional to the number of DARS loci. In addition, the drop of β_{datA} shown in Fig. S7B is not obvious to me either.”

Reply: We thank the referee for this key point. We have adopted a standard Michaelis-Menten description, in which the activities of *datA* and *DARS1/2* are proportional to their total copy number per unit volume times their occupancy (degree of saturation) by DnaA-ATP and DnaA-ADP, respectively (see section S3). Since *datA* is very close to the origin of replication, *oriC*, its copy number per unit volume is to a good approximation given by the *oriC* density. This description predicts that the rates are proportional to the effective copy numbers of *datA* and *DARS1/2*, respectively. In Fig. S16, we predict the effect of removing the *datA* and *DARS1/2* loci on the initiation volume in our full model, and showed that the prediction compares favorably with experiments. Yet, our model makes a much more non-trivial prediction: moving their loci on the chromosome can change replication initiation, via two effects: 1) moving a locus towards the terminus lowers its *effective, cell-cycle averaged* copy number; this causes the initiation volume to rise when *DARS2* is moved towards the terminus and to fall when *datA* is moved towards the terminus; interestingly, this effect becomes weaker at lower growth rates, because replication then happens for a shorter period during the cell cycle; 2) moving the locus towards the terminus can also change the temporal activity profile, which can not only lead to premature reinitiation events, but also counteract the first effect, as we describe in our response to comment 7 of referee 2, and in more detail in a new paragraph on *datA* in section S4B1, with a new figure Fig. S16J. As we describe here, our predictions are in agreement with the data of Frimodt-Møller et al. (PLoS Genetics, 2016) and Inoue et al. (Genes to Cells, 2016). This supports the idea that the (de)activation rate is described by a Michaelis-Menten model, in which the rate depends on the concentration of the locus and its occupancy by (in)active DnaA.

Concerning the comment in the last sentence: The temporal variations indicated in Fig. S7B models the activity of the protein IHF. Kasho et al., 2012 (doi.org/10.1073/pnas.1212070110) report that “[...] *IHF binds to datA immediately after initiation and that it dissociates from datA 20-30 min after initiation*”. We approximate this experimental observation by introducing a low and a high activity state of *datA*. We estimated the amplitude of these variations in IHF binding based on Fig. 4C of Kasho et al., 2012 to be approximately a factor of two, as discussed in section S3A.

Action: We have now added to section S4B1 the prediction of moving the *datA* locus, and added a new panel J to Fig. S16, showing how the initiation volume varies as the *datA* locus is moved. Moreover, we now compare the prediction with the data of Frimodt-Møller et al. and Inoue et al., not only on moving the locus of *DARS2* but also that of *datA*.

Comment 4: *“In the LD model, the authors state that the DDAH, rather than the RIDA, play the key role of DnaA-ATP deactivation. This choice makes the model highly dependent of the datA location. However, Frimodt et al. have reported that the changing of the datA location around the chromosome, even located nearby the ter site, have no significant effect on the initiation timing and initiation mass (Doi:10.1371/journal.pgen.1006286).”*

Reply: We carefully studied this paper, also because referee 2 noted this paper (comment 7). Moving the locus of *datA* to the terminus yields an origin density that is below wild-type. The authors write on p. 3 “*This suggests that the copy number is important for correct datA function.*” and in the Discussion, on p. 14 “*Thus, when datA is relocated to terC, ... gene dosage is diminished resulting in an increased DnaA-ATP / DnaA-ADP ratio and increased origin concentration.*” This is, in fact, precisely what our model predicts: moving *datA* towards the terminus lowers its effective copy number and hence its activity (see previous point, first effect), which decreases the initiation volume and increases the origin density. Prompted by the question of the referee and referee 2 (comment 7), we now demonstrate this explicitly, in a new panel added to Fig. S16, Fig. S16J.

Action: We have added the prediction of moving the *datA* locus to section S4B1 and added a new panel J to Fig. S16, showing how the initiation volume changes as the *datA* locus is moved.

REVIEWER COMMENTS

Reviewer #1 (Remarks to the Author):

The authors have adequately responded to the comments and I recommend publication.

Reviewer #2 (Remarks to the Author):

In this region, the authors added important explanations and modifications. Those sufficiently improved this manuscript.

Reviewer #3 (Remarks to the Author):

I appreciate the authors' efforts and in carefully responding the comments and revising the manuscript, which was significantly improved. Yet, my concerns haven't been fully addressed.

1. The authors ruled out the AIT model because of the re-initiation events at the intermediate growth regime. As I have stated in my earlier comments, it is not unlikely that the re-initiation phenomenon is parameter-dependent. However, the authors have not fully addressed this concern by providing compelling evidence through more extensive parameter tuning, despite their claim that "... this prediction of our model is robust, i.e. insensitive to the precise details of our model". Therefore, it is still not convincing that the AIT model can be disproved solely because of re-initiation.

Re-initiation occurrences have been simply attributed to the difference in the timescales of the doubling time and the C period. I agree that, in Figure 2B and Figure S3D, after the first initiation, the difference in the two timescales might explain why the cells initiated a new round of replication immediately when the eclipse period was over. However, this simple mechanism may not be sufficient to account for Figure 2C, where the premature initiation did not take place immediately following the eclipse period. These results imply that the dynamics of the re-initiation phenomenon is sensitive to parameters.

The authors analyzed the effects of inhibiting DnaA synthesis during the eclipse period on the occurrence of re-initiation. They found when the duration of the eclipse period was fixed at 10 min, re-initiation was prevented at high growth rates ($\lambda > 1.5 \text{ h}^{-1}$), but not at intermediate growth rates ($1 \text{ h}^{-1} < \lambda < 1.5 \text{ h}^{-1}$). In the main text, they argue that "... the effect of SeqA becomes weaker because of the fixed duration of the eclipse period". This raises the question of whether changing the duration of the eclipse period will change (or even broaden) the growth rate regime where re-initiation can be avoided. One reason for this point is that experiments have already shown that the eclipse period was not fixed but increased as the growth rate decreased (Table 3 of Ref. 52 in the manuscript). The fact that inhibiting DnaA synthesis during the eclipse period rescued the AIT model at high growth rates, again, indicates that the re-initiation issue could be parameter-sensitive and deserves more thorough analysis.

2. The authors stated that the lipid dynamics was not essential to the full model. However, without lipid dynamics, the full model becomes a trivial modification of the titration model. DNA replication generates new titration sites, leading to a decreasing number of free initiators. In the meantime, DNA replication activates the RIDA process and duplicates the *datA* site, both of which lead to the transformation of DnaA-ATP to DnaA-ADP. It is not surprising that adding the DDAH and RIDA mechanisms will increase the amplitude of the free-DnaA-ATP oscillations, and thus further stabilize the titration model.

3. Considering the points raised above, the novelty of this work largely lies in the LD/LDDR models.

The LD model is the foundation of the LDDR model. The LD model generated strong oscillations of DnaA-ATP and reproduced the adder phenomenon, which is non-trivial. However, in the SI text, the authors showed that in the absence of lipid-based DnaA activation, the amplitude of the oscillations becomes extremely small. Moreover, when lipid-mediated DnaA activation was taken out completely from the LD model, the authors did not clarify whether the adder phenomenon still remained (which I suppose not). Finally, it seems that the essence of the LD model depends on the hypothesis that the lipid dynamics activates DnaA. Since this hypothesis is still under debate, the foundation of the LD model becomes questionable.

Minor issues:

- Blue arrows on the top of Figure 2D should span more than two τ_d .
- On page 4, in "at intermediate growth rates ($1 > \lambda > 1.5 \text{ h}^{-1}$)", ">" should be "<".

Reply to Reviewers (Berger and Ten Wolde, *Robust replication initiation from coupled homeostatic mechanisms*)

We are pleased that reviewers 1 and 2 are satisfied with our response and have now recommended publication of our manuscript. Below we respond to the comments of reviewer 3 point by point. The changes made to the main text and the SI are highlighted in blue.

Response to reviewer #3

Comment: *“I appreciate the authors efforts and in carefully responding the comments and revising the manuscript, which was significantly improved.”*

Reply: We are pleased that the referee believes our manuscript has significantly improved.

Comment 1: *“The authors ruled out the AIT model because of the re-initiation events at the intermediate growth regime. As I have stated in my earlier comments, it is not unlikely that the re-initiation phenomenon is parameter-dependent. However, the authors have not fully addressed this concern by providing compelling evidence through more extensive parameter tuning, despite their claim that “... this prediction of our model is robust, i.e. insensitive to the precise details of our model”. Therefore, it is still not convincing that the AIT model can be disproved solely because of re-initiation.”*

“Re-initiation occurrences have been simply attributed to the difference in the timescales of the doubling time and the C period. I agree that, in Figure 2B and Figure S3D, after the first initiation, the difference in the two timescales might explain why the cells initiated a new round of replication immediately when the eclipse period was over. However, this simple mechanism may not be sufficient to account for Figure 2C, where the premature initiation did not take place immediately following the eclipse period. These results imply that the dynamics of the re-initiation phenomenon is sensitive to parameters.”

Reply: We appreciate the question of the referee to analyze in even more detail the breakdown of the titration-based mechanism, because we consider this to be one of the strongest predictions of our modelling. Here, it is important to make a distinction between the behavior of a system based on titration alone and that of a system that combines titration with the periodic suppression of DnaA synthesis by SeqA.

Starting with the titration-only model: the titration mechanism breaks down because of the different scaling of the protein synthesis rate and the titration-site formation rate with the growth rate. As we wrote in the SI (section S2B4), the rate at which new titration sites are formed during replication is $dN_s/dt = n_s(n_{\text{ori}} - 1)/T_C$, while the rate at which DnaA proteins are synthesized is given by $dN_p/dt = \lambda N_p = \ln(2)N_p/\tau_d$. In the non-overlapping fork regime, just after replication initiation, $n_{\text{ori}} = 2$ and $N_p \simeq n_s$, such that the protein synthesis rate is $dN_p/dt \simeq \ln(2)n_s/\tau_d$. Hence, the protein synthesis rate becomes higher than the titration-site formation rate when $\tau_d \lesssim T_C$ and the system transitions into the overlapping fork regime. We now provide more support for this argument by considering replication mutants in which T_C is altered: as the new figure S3E shows, the titration mechanism breaks down when the system enters the overlapping fork regime, even though in these mutants this regime starts at other growth rates. This figure clearly shows that

the breakdown of the titration mechanism is rooted in the different scaling of the DnaA synthesis and titration-site formation rate with the growth rate.

The next question is then how the periodic suppression of DnaA synthesis by SeqA changes the behavior of the system. Adding periodic suppression of DnaA synthesis by SeqA to the titration model does indeed eliminate *immediate* reinitiation events as soon as the blocked period is over, but, as Fig. 2C shows, it still leads to premature reinitiation events: while in stable replication cycles, as occur in the non-overlapping fork regime, the time between successive initiation events is constant and precisely equal to the cell doubling time τ_d , in the intermediate growth rate regime shown in Fig. 2C, the time between successive initiation events alternates between a time that is shorter than τ_d and one that is longer than τ_d ; this succession of alternating replication initiation cycles gives rise to variations in the initiation volume, even though the system is deterministic. Crucially, the origin of this is precisely the mechanism shown in Fig. 2B, namely that in the overlapping fork regime proteins are produced faster than titration sites per origin are.

Action: We have now changed the main text to discuss this more clearly and we have also added a new figure S3E to the SI where we show that in the model based on titration only, premature reinitiation events happen when the system enters the overlapping fork regime.

Comment: *“The authors analyzed the effects of inhibiting DnaA synthesis during the eclipse period on the occurrence of re-initiation. They found when the duration of the eclipse period was fixed at 10 min, re-initiation was prevented at high growth rates ($\lambda > 1.5\text{h}^{-1}$), but not at intermediate growth rates ($1\text{h}^{-1} < \lambda < 1.5\text{h}^{-1}$). In the main text, they argue that “... the effect of SeqA becomes weaker because of the fixed duration of the eclipse period”. This raises the question of whether changing the duration of the eclipse period will change (or even broaden) the growth rate regime where re-initiation can be avoided. One reason for this point is that experiments have already shown that the eclipse period was not fixed but increased as the growth rate decreased (Table 3 of Ref. 52 in the manuscript). The fact that inhibiting DnaA synthesis during the eclipse period rescued the AIT model at high growth rates, again, indicates that the re-initiation issue could be parameter-sensitive and deserves more thorough analysis.”*

Reply: We thank the referee for this interesting point. In the revised manuscript, we now also study systematically the effect of varying the duration of the eclipse period at different growth rates. In Fig. S4, we show the coefficient of variation (CV) of the initiation volume in a deterministic model for a broad range of growth rates λ and durations of the eclipse period τ_b . We also marked the in Ref. 52 reported durations of the eclipse period at the respective growth rates (white dots). The figure shows that in the non-overlapping fork regime the CV is indeed zero, but rises in the overlapping fork regime. The figure also nicely shows that increasing the duration τ_b of the eclipse period helps to prevent the rise of the CV, but not sufficiently: at intermediate growth rates ($1\text{h}^{-1} < \lambda < 1.5\text{h}^{-1}$), the CV is still higher than that reported experimentally when the duration of the eclipse period is in the range reported experimentally ($\tau_b \sim 4 - 10\text{min}$); in fact, the CV is higher even though the model is deterministic and other noise sources, which would only raise the CV, are excluded. In this regime of intermediate growth rates, another mechanism—the activation switch—becomes essential to generate stable replication cycles.

Action: We have added an extra figure S4 to the SI, which shows the CV of the initiation volume as a function of the growth rate and duration of the eclipse period. We discuss the figure in a new SI section, S2B5, and mention it in the main text.

Comment 2: *“The authors stated that the lipid dynamics was not essential to the full model. However, without lipid dynamics, the full model becomes a trivial modification of the titration model. DNA replication generates new titration sites, leading to a decreasing number of free initiators. In the meantime, DNA replication activates the RIDA process and duplicates the *datA* site, both of which lead to the transformation of DnaA-ATP to DnaA-ADP. It is not surprising that adding the DDAH and RIDA mechanisms will increase the amplitude of the free-DnaA-ATP oscillations, and thus further stabilize the titration model.”*

Reply and Action: A few points are worthy of note. First of all, the full model consists of two distinct cycles: a titration-driven cycle in the total concentration of cytoplasmic DnaA (be it active or inactive), and a cycle in the fraction of active DnaA, as driven by the nucleotide switch. Even without the lipids, there is still an activation cycle driven by DDAH, RIDA and *DARS1/2*. It is important to realize that there are two distinct cycles, because it raises the question how these cycles interact. As we know from previous work, including our own work on the cyanobacterial clock (Zwicker et al, PNAS, 2010; Paijmans et al., PNAS, 2016), stability often arises from the interaction between the respective cycles. This observation is particularly relevant here, because a system built on only a titration-based concentration cycle breaks down in the high-growth rate regime of overlapping replication forks, while a system based on only an activation cycle driven by DDAH, RIDA, and *DARS1/2* (i.e. without the lipids) is unstable at all growth rates; only the system that combines these cycles is stable at all growth rates, which is a non-trivial observation. Indeed, the titration-based concentration cycle requires the help from the activation cycle, be it with or without the lipids: the activation cycle does not merely *“further stabilize the titration model”*, but is essential to generate stable rhythms at all growth rates. Conversely, an activation cycle without the lipids needs a titration-based concentration cycle to become stable. With lipid-mediated DnaA rejuvenation, the activation cycle is stable by itself at all growth rates, but a concentration cycle driven by titration can still help the activation cycle by making the latter more robust against fluctuations in the switch components. We thus find that the interaction between the two respective cycles is vital to the stability of the system, which we now emphasize more clearly in the Discussion section.

Comment 3: *“Considering the points raised above, the novelty of this work largely lies in the LD/LDDR models. The LD model is the foundation of the LDDR model. The LD model generated strong oscillations of DnaA-ATP and reproduced the adder phenomenon, which is non-trivial. However, in the SI text, the authors showed that in the absence of lipid-based DnaA activation, the amplitude of the oscillations becomes extremely small. Moreover, when lipid-mediated DnaA activation was taken out completely from the LD model, the authors did not clarify whether the adder phenomenon still remained (which I suppose not). Finally, it seems that the essence of the LD model depends on the hypothesis that the lipid dynamics activates DnaA. Since this hypothesis is still under debate, the foundation of the LD model becomes questionable.”*

Reply and Action: The referee seems to raise the question what the novelty and importance of our results are given the conflicting evidence on the role of the lipids in DnaA rejuvenation. We therefore here summarize the key findings of our work, and how they depend on the role of the lipids. 1) a model built on titration alone inevitably breaks down in the regime of overlapping replication forks; 2) the periodic suppression of DnaA synthesis can rescue the titration mechanism, but only at high growth rates. 3) A third mechanism is therefore essential; according to our model, this is the activation cycle of DnaA driven by the nucleotide switch. Importantly, none of these three predictions rely on lipid-mediated DnaA rejuvenation. Also the third one does not: an activation cycle is able to interact with a titration-based concentration cycle to generate stable replication cycles at all growth rates, *with or without the lipids*. The fact that the importance of the activation cycle in driving robust rhythms in concert with titration does not critically depend on the role of the lipids, is only a strength of our model. At the same time, we believe it is an interesting prediction from our model that while an activation cycle with lipids can be stable by itself, an activation cycle without lipids requires the help of the titration-based concentration cycle. It shows that the question of homeostasis and stability is non-trivial.

Given the evidence that lipids enhance the rejuvenation of DnaA, we feel it is natural to start the switch section of the main text with the LD model, also because it is the simplest model to elucidate the design logic of the nucleotide switch. A lipid-devoid switch is not stable by itself and the question whether it exhibits adder correlations is therefore moot, but we have performed additional simulations which show that the full model without the lipids does exhibit adder correlations (new panel Fig. S21H, mentioned in Discussion).

Finally, we would like to emphasize that the importance of our modelling lies in the combination of predictions 1-3, because only together do they explain why the system combines these different mechanisms. It is also the full model that can explain a large body of experimental observations, ranging from experiments that test the role of titration by varying the total DnaA concentration and number of titration sites, to experiments that probe the importance of DnaA activation using mutants.

Comment: *“Minor issues:*

- *Blue arrows on the top of Figure 2D should span more than two τ_d .*
- *On page 4, in “at intermediate growth rates ($1 > \lambda > 1.5h^{-1}$)”, $>$ should be $<$.”*

Reply and Action: We would like to thank the referee very much for catching these errors, which we have corrected in the revised manuscript.

REVIEWERS' COMMENTS

Reviewer #3 (Remarks to the Author):

The authors have addressed all my comments. I suggest the manuscript to be accepted for publication.

Reply to Reviewers (Berger and Ten Wolde, *Robust replication initiation from coupled homeostatic mechanisms*)

Response to reviewer #3

Comment: *“The authors have addressed all my comments. I suggest the manuscript to be accepted for publication.”*

Reply: We are pleased that reviewer 3 is satisfied with our response and has now recommended publication of our manuscript.